# NEURAL-SYMBOLIC MESSAGE PASSING WITH DYNAMIC PRUNING

## ABSTRACT

Complex Query Answering (CQA) over Knowledge Graphs (KGs) is a fundamental yet challenging task. Given that KGs are usually incomplete, the CQA models not only need to execute logical operators, but aslo need to leverage observed knowledge to predict the missing one. Recently, a line of message-passing-based research has been proposed to re-use pre-trained neural link predictors to solve CQA. However, they perform unsatisfactorily on negative queries and fail to address the unnecessary noisy messages between variable nodes in the query graph. Moreover, like most neural CQA models, these message passing models offer little interpretability and require complex query data and resource-intensive training. In this paper, we propose a Neural-Symbolic Message Passing framework (NSMP) based on pre-trained neural link predictors. By introducing symbolic reasoning and fuzzy logic, NSMP can generalize to arbitrary existential first order logic queries without requiring training on any complex queries while providing interpretable answers. Furthermore, we introduce an effective dynamic pruning strategy to filter out noisy messages between variable nodes during message passing. Empirically, our model demonstrates strong performance and offers efficient inference. Our code can be found at https://anonymous.4open.science/r/NSMP.

## 1 INTRODUCTION

Knowledge graphs (KGs) store factual knowledge in the form of graph representations, which can be applied to various intelligent application scenarios (Saxena et al., 2020; Wang et al., 2021a). Complex Query Answering (CQA) over KGs is a fundamental and practical task, which requires answering existential first order logic formula with logical operators including conjunction ($\wedge$), disjunction ($\vee$), negation ($\neg$), and existential quantifier ($\exists$). A straightforward way is to traverse the KG to identify the answers directly (Zou et al., 2011). However, given that modern KGs are usually auto-generated (Toutanova & Chen, 2015) or built through crowd-sourcing (Vrandečić & Krötzsch, 2014), real-world KGs (Bollacker et al., 2008; Carlson et al., 2010) often suffer from incompleteness, which is also known as the Open World Assumption (OWA) (Libkin & Sirangelo, 2009; Ji et al., 2021). This makes it impossible to answer a complex query with missing links using traditional traversal methods. Therefore, the CQA models not only need to execute logical operators, but also need to utilize available knowledge to predict the unseen one (Ren & Leskovec, 2020).

Inspired by the success of neural link predictors (Bordes et al., 2013; Trouillon et al., 2016; Sun et al., 2019; Li et al., 2022) on answering one-hop atomic queries on incomplete KGs, neural models (Hamilton et al., 2018; Ren & Leskovec, 2020; Zhang et al., 2021; Chen et al., 2022; Wang et al., 2023b; Zhang et al., 2024b) have been proposed to represent the entity sets by low-dimensional embeddings. Building on the foundation laid by these neural CQA models, message-passing-based research (Wang et al., 2023c; Zhang et al., 2024a) has demonstrated promising advancements in CQA. These message passing approaches represent logical formulas as query graphs, where each edge represents an atomic formula containing a predicate with a (possible) negation operator, and each node represents an input constant entity or a variable, as illustrated in Figure 1. By utilizing pre-trained neural link predictors, they perform one-hop inference on atomic formulas, thereby inferring intermediate embeddings for variable nodes. An intermediate embedding is interpreted as a logical message passed from the neighboring node on the corresponding edge. Following the message passing paradigm (Gilmer et al., 2017), the embeddings of variable nodes are updated to retrieve answers. Due to the integration of pre-trained neural link predictors, these message pass-

ing approaches are effective on both one-hop atomic and multi-hop complex queries. However, limitations still exist.

Firstly, even when augmented with fuzzy logic (Hájek, 2013) for one-hop inference on atomic formulas, existing message passing CQA models still perform unsatisfactorily on negative queries.

Secondly, while recent work (Zhang et al., 2024a) has considered noisy messages between variable and constant nodes, noisy messages between variable nodes remain unexplored. At the initial layers of message passing, messages inferred from neighboring variable nodes—whose states have not yet been updated—are ineffective. Aggregating such messages to update node states is essentially equivalent to introducing noise. Thirdly, similar to most neural CQA models, they offer little interpretability and require training on large complex query datasets, which entails substantial training costs. In practical scenarios, gathering meaningful complex query data poses a significant challenge.

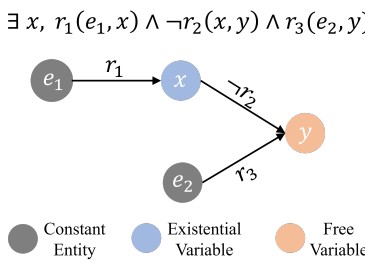

Figure 1: A query graph representation of a given logical formula.

In this paper, we propose a Neural-Symbolic Message Passing framework (NSMP), which leverages a simple pre-trained neural link predictor without requiring training on any complex query data. Specifically, NSMP conducts one-hop inference on atomic formulas by integrating neural and symbolic reasoning to compute intermediate states for variable nodes. The intermediate state can be interpreted as a message, represented by a fuzzy vector that denotes the fuzzy set of the variable within the corresponding atomic formula. In particular, we propose a novel pruning strategy that dynamically filters out unnecessary noisy messages between variable nodes during message passing. Based on fuzzy logic theory, NSMP aggregates the messages received by the variable nodes and updates the node states. Such a mechanism attains interpretability for the variables in the formula and can naturally execute the negation operator through fuzzy logic. Extensive experiments results show that NSMP achieves competitive performance with more efficient inference. In general, our main contributions can be summarized as follows:

- We propose a neural-symbolic message passing framework that, for the first time, integrates neural and symbolic reasoning within a message passing CQA model. By leveraging fuzzy logic theory, our approach can naturally answer arbitrary existential first order logic formulas without requiring training on complex queries while providing interpretability.

- We propose a novel dynamic pruning strategy to filter out unnecessary noisy messages between variable nodes during message passing, thereby reducing unnecessary computation and improving performance.

- Extensive experimental results on benchmark datasets show that NSMP achieves a strong performance. In particular, NSMP significantly improves the performance of message passing CQA models on negative queries by introducing symbolic reasoning and fuzzy logic.

- Through computational complexity analyses, we reveal that our message-passing-based method can provide more efficient inference than the current state-of-the-art step-by-step neural-symbolic method and empirically verify this.

## 2 RELATED WORK

In recent years, neural models (Hamilton et al., 2018; Ren & Leskovec, 2020; Zhang et al., 2021; Amayuelas et al., 2022) have been proposed to solve complex query answering by representing sets of entities using low-dimensional embeddings. Among these models, message-passing-based approaches (Wang et al., 2023c; Zhang et al., 2024a) have demonstrated promising potential. However, they overlook noisy messages between variables and, compared to symbolic integration methods (Arakelyan et al., 2021; Xu et al., 2022; Bai et al., 2023b), suffer from reliance on training data and a lack of interpretability. In contrast, our proposed message-passing approach can effectively address these challenges. Recently, Yin et al. (2024) proposed a step-by-step neural-symbolic method, which achieves state-of-the-art performance but suffers from inefficiency. In contrast, our message-

passing-based NSMP provides more efficient inference while achieving competitive performance. Further discussion of related work can be found in Appendix A.

# 3 BACKGROUND

## 3.1 KNOWLEDGE GRAPHS

Given a set of entities $\mathcal{V}$ and a set of relations $\mathcal{R}$, a knowledge graph $\mathcal{KG}$ can be defined as a set of triples $\mathcal{E} = \{(e_{h_i}, r_i, e_{t_i})\}$ that encapsulates the factual knowledge, where each triple encodes a relationship of type $r_i \in \mathcal{R}$ between the head and tail entity $e_{h_i}, e_{t_i} \in \mathcal{V}$. According to the definition in Wang et al. (2022b; 2023c), the $\mathcal{KG}$ is an $\mathcal{L}_{\mathcal{KG}}$-structure defined by a language $\mathcal{L}_{\mathcal{KG}}$. Specifically, a first-order language $\mathcal{L}$ can be defined as a triple $(\mathcal{F}, \mathcal{R}, \mathcal{C})$, where $\mathcal{F}$, $\mathcal{R}$, and $\mathcal{C}$ are sets of function symbols, predicate symbols, and constants symbols, respectively. Under language $\mathcal{L}_{\mathcal{KG}}$, the predicate symbols in $\mathcal{R}$ denote binary relations, the constant symbol set $\mathcal{C}$ is finite and comprises all entities, and the set of function symbol satisfies $\mathcal{F} = \emptyset$. In this case, each entity $e \in \mathcal{V}$ is also a constant $c \in \mathcal{C} = \mathcal{V}$ and each relation $r \in \mathcal{R}$ is a set $r \subseteq \mathcal{V} \times \mathcal{V}$. In addition, only a part of the complete knowledge graph $\mathcal{KG}$ can be observed due to OWA. Let $\mathcal{KG}_{obs}$ be the observed knowledge graph, we have $\mathcal{KG}_{obs} \subsetneq \mathcal{KG}$.

## 3.2 EXISTENTIAL FIRST ORDER QUERIES WITH A SINGLE FREE VARIABLE

The complex queries that existing studies aim to address exclude universal quantifiers (Ren et al., 2020; Ren & Leskovec, 2020). We follow Yin et al. (2024) and formally define such queries as Existential First Order queries with a single free variable (EFO$_1$), which are an important subset of first-order queries, using existential quantifier, conjunction, disjunction, and atomic negation. Following (Marker, 2006; Wang et al., 2022b; Yin et al., 2024), we give a set of definitions to describe the EFO$_1$ queries over knowledge graphs.

**Definition 1.** (Atomic Formula). *An atomic formula is of the form $\phi = r(t_1, t_2)$, where $r \in \mathcal{R}$ and each $t_i$ is a term that is either a variable or a constant $c \in \mathcal{C} = \mathcal{V}$.*

**Definition 2.** (Existential First Order Formula). *The set of existential formulas is the smallest set $\Phi$ that satisfies the following property:*

- *An atomic formula and its negation $\phi = r(t_1, t_2), \neg\phi = r(t_1, t_2) \in \Phi$;*

- *If $\phi, \psi \in \Phi$, then $(\phi \wedge \psi), (\phi \vee \psi) \in \Phi$;*

- *If $\phi \in \Phi$ and $x_i$ is any variable, then $\exists x_i \phi \in \Phi$.*

A variable is a bound variable when associated with a quantifier. Otherwise, it is a free variable. For an existential first order formula $\phi$ containing a free variable $y$, that is, an EFO$_1$ formula, we denote it as $\phi(y)$. Specifically, a formula with at least one free variable can be referred to as a query, whereas one with no free variables can be called a sentence.

**Definition 3.** (Substitution). *Given an EFO$_1$ formula $\phi(y)$, for any constant entity $e \in \mathcal{V}$, we denote $\phi(e)$ as the result of replacing all free occurrences of $y$ in $\phi$ with $e$.*

**Definition 4.** (The Answer Set of EFO$_1$ Query). *For an EFO$_1$ query $\phi(y)$, its answer set $\mathcal{A}[\phi(y)] \subseteq \mathcal{V}$ is a set of entities such that $e \in \mathcal{A}[\phi(y)]$ iff $\phi(e) = True$.*

Most previous studies convert EFO$_1$ queries into Disjunctive Normal Form (DNF) (Davey & Priestley, 2002) to handle disjunction operators in a scalable manner (Ren et al., 2020; 2022; Wang et al., 2023b; Zhang et al., 2024a). Our work can also apply such DNF-based processing to answer complex queries. Therefore, we also give the definition of DNF here.

**Definition 5.** (Disjunctive Normal Form). *The disjunctive normal form $\phi_{\mathrm{DNF}}$ of an EFO$_1$ formula is*

$$\phi_{\mathrm{DNF}}(y) = CF_1(y) \vee ... \vee CF_d(y), \tag{1}$$

*where $CF_i = \exists x_1, ..., \exists x_k. a_1^i \wedge ... \wedge a_{n_i}^i$ is a conjunctive formula, $x_1, ..., x_k$ are existential variables, $y$ is the only free variable, and $a_j^i$ are atomic formulas or its negation.*

According to Ren et al. (2020), the answer set $\mathcal{A}[\phi_{\mathrm{DNF}}(y)]$ can be obtained by taking the union of the answer sets of each sub-conjunctive formula, i.e., $\mathcal{A}[\phi_{\mathrm{DNF}}(y)] = \bigcup_{i=1}^{d} \mathcal{A}[CF_i(y)]$. This means that solving all conjunctive formulas $CF_i, 1 \leq i \leq d$ yields the answer set $\mathcal{A}[\phi_{\mathrm{DNF}}(y)]$ for the $\mathrm{EFO}_1$ query in DNF.

### 3.3 Query Graph

As mentioned above, since a DNF query can be addressed by solving all its sub-conjunctive queries, it is only necessary to define query graphs for conjunctive formulas. Following Wang et al. (2023c); Zhang et al. (2024a); Yin et al. (2024), we represent the conjunctive formula as the query graph where the terms are represented as nodes connected by the atomic formulas.

**Definition 6.** (Query Graph). *For a conjunctive formula $CF$, its query graph $G(CF)$ is defined as $\{(h, r, t, \{0/1\})\}$, where each quadruple represents an atomic formula and specifies whether it is negated. The quadruple defines an edge with two endpoints $h, t$, and two attributes: $r$ and 0/1. Here, $r$ denotes the relation, and 0/1 indicates whether the atomic formula is positive or negated.*

According to Definition 1, each node in the query graph is either a constant entity or a free or existential variable, as illustrated in the example in Figure 1.

### 3.4 Neural Link Predictors

A neural link predictor is a differentiable model that embeds entities and relations into a low-dimensional vector space to produce confidence scores for triples. Specifically, let $h$, $r$ and $t$ represent the embedding of the head entity, relation, and tail entity, respectively. The neural link predictor uses its scoring function to compute the likelihood that a given triple exists. In particular, by applying a sigmoid function $\sigma$, it can output a continuous truth value $\varphi(h, r, t) \in [0, 1]$ for the triple. For example, we can get the truth value for a triple through ComplEx (Trouillon et al., 2016):

$$\varphi(h, r, t) = \sigma(Re\left(\langle h \otimes r, \bar{t} \rangle\right)), \tag{2}$$

where $\otimes$ denotes element-wise complex number multiplication, $\langle \cdot, \cdot \rangle$ represents the complex inner product, and $Re$ refers to extracting the real part of a complex number. Similar to other message-passing-based models (Wang et al., 2023c; Zhang et al., 2024a), the input to the neural link predictor in our work includes not only embeddings of specific entities but also embeddings of variables. Following previous works (Arakelyan et al., 2021; 2023; Wang et al., 2023c), in our work, we use ComplEx-N3 (Trouillon et al., 2016; Lacroix et al., 2018) as the neural link predictor of NSMP.

### 3.5 Neural One-hop Inference on Atomic Formulas

According to Definition 6, each edge in a query graph represents either an atomic formula or a negated atomic formula. As such, an edge encodes information about the relation, logical negation, and terms. Previous message passing CQA models (Wang et al., 2023c; Zhang et al., 2024a) leverage this information to perform neural one-hop inference that maximizes the continuous truth value of the (negated) atomic formulas, computing intermediate embeddings (i.e., logical messages) for the variable nodes in the query graph. Specifically, a logical message encoding function $\rho$ is proposed to define this neural one-hop inference on edges. On each edge, when a node is at the head position, its neighbor is at the tail position, and vice versa. The function $\rho$ takes four input arguments: the neighbor embedding, the relation embedding, the direction information (either $h \rightarrow t$ or $t \rightarrow h$), and the logical negation indicator (0 for no negation and 1 for with negation). Depending on these input arguments, $\rho$ operates in four distinct cases. Given the tail embedding $t$ and relation embedding $r$ on a non-negated atomic formula, $\rho$ is formulated in the form of continuous truth value maximization to infer the intermediate embedding $\hat{h}$ for the node at the head position on this edge:

$$\hat{h} = \rho(t, r, t \rightarrow h, 0) := \underset{x \in \mathcal{D}}{arg\,max}\; \varphi(x, r, t), \tag{3}$$

where $\mathcal{D}$ is the search domain for the embedding $x$. Similarly, the intermediate embedding $\hat{t}$ for the node at the tail position on a non-negated edge can be inferred given $h$ and $r$:

$$\hat{t} = \rho(h, r, h \rightarrow t, 0) := \underset{x \in \mathcal{D}}{arg\,max}\; \varphi(h, r, x). \tag{4}$$

Based on the fuzzy logic negator (Hájek, 2013), the estimation of intermediate embeddings on negated atomic formulas can be defined as follows:

$$\hat{h} = \rho(t, r, t \to h, 1) := \underset{x \in \mathcal{D}}{arg\, max}\, \varphi(x, \neg r, t) = \underset{x \in \mathcal{D}}{arg\, max}\, [1 - \varphi(x, r, t)], \tag{5}$$

$$\hat{t} = \rho(h, r, h \to t, 1) := \underset{x \in \mathcal{D}}{arg\, max}\, \varphi(h, \neg r, x) = \underset{x \in \mathcal{D}}{arg\, max}\, [1 - \varphi(h, r, x)]. \tag{6}$$

In our work, we perform symbolic-integrated one-hop inference on edges, incorporating both neural and symbolic components into our message encoding function. We use the above neural message encoding function $\rho$ as the neural component of our neural-symbolic message encoding function.

## 4 PROPOSED METHOD

In this section, we first propose how to integrate symbolic reasoning into neural one-hop inference to compute neural-symbolic messages. Then, we propose our message passing mechanism based on fuzzy logic and dynamic pruning. Finally, we analyze the computational complexity of the proposed method to reveal the superiority of the message-passing-based method in terms of efficiency.

### 4.1 NEURAL-SYMBOLIC ONE-HOP INFERENCE ON ATOMIC FORMULAS

To integrate symbolic information into neural message passing, there are two different representations of entities and relations in our work: neural and symbolic representations. For the neural representation, since we utilize the pre-trained neural link predictor to compute neural messages, the entities and relations are encoded into the embedding space of the pre-trained neural link predictor. That is, they already have pre-trained embeddings. For the symbolic representation, each entity $e \in \mathcal{V}$ is encoded as a one-hot vector $p_e \in \{0, 1\}^{1 \times |\mathcal{V}|}$ and each relation $r \in \mathcal{R}$ is represented as an adjacency matrix $M_r \in \{0, 1\}^{|\mathcal{V}| \times |\mathcal{V}|}$, where $M_r^{ij} = 1$ if $(e_i, r, e_j) \in \mathcal{E}$ else $M_r^{ij} = 0$. We follow Zhang et al. (2024a) and only consider inferring the intermediate state for the variable, so the neural and symbolic representations of entities and relations remain unchanged. Accordingly, we also assign neural and symbolic representations to the variable nodes in the query graph. Specifically, each variable is equipped with a corresponding embedding and symbolic vector. However, the symbolic representation of a variable is not a one-hot vector but a fuzzy vector $p_v \in [0, 1]^{1 \times |\mathcal{V}|}$ that represents a fuzzy set. Each element of $p_v$ can be interpreted as the probability of the corresponding entity.

To conduct symbolic reasoning on atomic formulas, we follow TensorLog (Cohen et al., 2020) and define a symbolic one-hop inference function $\mu$ for four cases depending on the input arguments. The first situation is to infer the tail symbolic vector $p_t$ given the head symbolic vector $p_h$ and relational adjacency matrix $M_r$ on a non-negated atomic formula:

$$p_t = \mu(p_h, M_r, h \to t, 0) := \mathcal{N}(p_h M_r), \tag{7}$$

where $\mathcal{N}(\triangle) = \triangle / sum(\triangle)$ is a normalized function. Similarly, the head symbolic vector $p_h$ can be inferred given $p_t$ and $M_r$:

$$p_h = \mu(p_t, M_r, t \to h, 0) := \mathcal{N}(p_t M_r^\top), \tag{8}$$

where $\top$ stands for transpose. Based on the fuzzy logic theory (Klement et al., 2013; Hájek, 2013), we follow Xu et al. (2022) and define the estimation of symbolic vector on negated atomic formulas as follows:

$$p_t = \mu(p_h, M_r, h \to t, 1) := \mathcal{N}(\frac{\alpha}{|\mathcal{V}|} - p_h M_r), \tag{9}$$

$$p_h = \mu(p_t, M_r, t \to h, 1) := \mathcal{N}(\frac{\alpha}{|\mathcal{V}|} - p_t M_r^\top), \tag{10}$$

where $\alpha$ is a hyperparameter. In order to integrate neural reasoning to enhance symbolic reasoning, we consider converting the intermediate embedding obtained by the neural message encoding function $\rho$ into a fuzzy vector. Specifically, for an intermediate embedding inferred by $\rho$, we first compute its similarity with the embeddings of all entities. After applying a softmax operation, we can obtain a fuzzy vector $p' \in [0, 1]^{1 \times |\mathcal{V}|}$. We define this procedure as a function $f$ as follows:

$$f(\rho) = softmax(\underset{\forall e \in \mathcal{V}}{concat}(\mathcal{S}(\rho, E_e))), \tag{11}$$

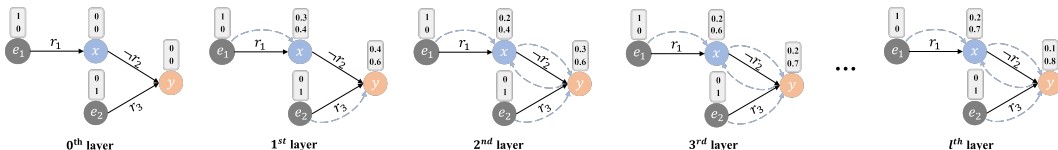

Figure 2: A toy example to show the process of dynamic pruning. The query graph follows Figure 1. The blue arrow represents the passing of the message computed by encoding function $\varrho$, and the vector indicates the state of the corresponding node after message passing at each layer.

where $E_e$ represents the embedding of the entity $e$, $\mathcal{S}$ is a binary similarity function, and *concat* is a function that maps the similarities between all entities and the intermediate embedding inferred by $\rho$ to a vector. Depending on the selected pre-trained neural link predictor, $\mathcal{S}$ can either be an inner-product-based or a distance-based scoring function. Then, we can define our neural-symbolic message encoding function $\varrho$, which also has four cases depending on the input arguments.

Given the embedding $t$ and symbolic vector $p_t$ of the tail term on a non-negated atomic formula, as well as the embedding $r$ and adjacency matrix $M_r$ of the relation, we infer the intermediate state $p_{\hat{h}}$ for the variable node at the head position on this edge. We formulate the inference task in the form of *neural-enhanced symbolic reasoning*:

$$p_{\hat{h}} = \varrho(t, p_t, r, M_r, t \to h, 0) := \mathcal{N}(f(\rho(t, r, t \to h, 0)) + \mu(p_t, M_r, t \to h, 0)). \quad (12)$$

This approach leverages the embeddings inferred by pre-trained neural link predictors to enhance symbolic reasoning results. It enables symbolic reasoning to handle missing links in observed knowledge graphs. Additionally, such an approach can represent the membership degree of variables concerning all entities in the form of fuzzy sets, thereby providing interpretability. Similarly, for the other three cases, the encoding functions $\varrho$ are as follows:

$$p_{\hat{t}} = \varrho(h, p_h, r, M_r, h \to t, 0) := \mathcal{N}(f(\rho(h, r, h \to t, 0)) + \mu(p_h, M_r, h \to t, 0)), \quad (13)$$
$$p_{\hat{h}} = \varrho(t, p_t, r, M_r, t \to h, 1) := \mathcal{N}(f(\rho(t, r, t \to h, 1)) + \mu(p_t, M_r, t \to h, 1)), \quad (14)$$
$$p_{\hat{t}} = \varrho(h, p_h, r, M_r, h \to t, 1) := \mathcal{N}(f(\rho(h, r, h \to t, 1)) + \mu(p_h, M_r, h \to t, 1)). \quad (15)$$

## 4.2 Neural-Symbolic Message Passing

In this subsection, we propose a Neural-Symbolic Message Passing framework (NSMP). This framework builds on the neural-symbolic one-hop inference proposed in Section 4.1 and incorporates a dynamic pruning strategy. As a variation of the message passing neural network (Gilmer et al., 2017), each NSMP layer has two stages: (1) Passing neural-symbolic messages based on our dynamic pruning strategy; (2) Updating the state of the variable node that has received messages.

### 4.2.1 Message Passing with Dynamic Pruning

A query graph contains two types of nodes: constant and variable nodes. For constant nodes, we follow Zhang et al. (2024a) and do not pass messages to constant nodes. However, the situation varies across different layers of message passing when passing messages to variable nodes. At the initial layers, messages passed from neighboring variable nodes, whose states have not yet been updated, can be regarded as noise, as these nodes do not carry any meaningful information at this stage. In contrast, at the later layers, messages from neighboring variable nodes with updated states provide valuable information derived from constants, which is especially important for variable nodes not directly connected to constant nodes. To dynamically filter out unnecessary noisy messages while retaining valuable ones, we decide whether a variable node should pass messages to its neighboring variable nodes based on whether its state has been updated. Specifically, a variable node is allowed to pass messages to its neighboring variable nodes only when its state has been updated. This is a dynamic pruning process, which can effectively avoid the computation of noise messages during message passing. Figure 2 illustrates an example of this pruning strategy at different layers.

### 4.2.2 NODE STATE UPDATE SCHEME

Let $z_e^{(l)}$ and $s_e^{(l)}$ denote the embedding and symbolic vector of a constant entity node at the $l^{th}$ layer, respectively, while $z_v^{(l)}$ and $s_v^{(l)}$ represent the embedding and symbolic vector of a variable node at the same layer. Next, we discuss how to compute these representations from the input layer $l = 0$ to latent layers $l > 0$. For a constant entity node, $z_e^{(0)}$ is the corresponding frozen entity embedding in the pre-trained neural link predictor, $s_e^{(0)}$ is the corresponding one-hot symbolic vector. For a variable node, both $z_v^{(0)}$ and $s_v^{(0)}$ are vectors containing all zeros. Since we do not pass messages to constant nodes, the embeddings and symbolic vectors of the constant nodes are the same at each layer, i.e., $z_e^{(l)} = z_e^{(0)}$ and $s_e^{(l)} = s_e^{(0)}$. While for $z_v^{(l)}$ and $s_v^{(l)}$, similar to the dynamic pruning proposed in Section 4.2.1, we only update the state of variable nodes that have received messages. That is, if the corresponding variable node receives messages from neighboring nodes, we utilize these messages to update the state of this node. Otherwise, we do not update its state at $l^{th}$ layer.

For a variable node, the messages it receives from neighboring nodes essentially represent intermediate states inferred by the neural-symbolic message encoding function $\varrho$, as proposed in Section 4.1. These intermediate states, which are fuzzy vectors representing fuzzy sets, can be aggregated using fuzzy logic theory to update the state of the variable node. Specifically, for a variable node $v$, we form a neighbor set $N_{DP}(v)$ that includes all its neighboring variable nodes with updated states and all its neighboring constant nodes, in accordance with the dynamic pruning strategy. For each neighboring node $n \in N_{DP}(v)$, one can obtain information about the edge between $n$ and $v$, which contains the neighbor mebdding $z_n^{(l-1)}$, the neighbor symbolic vector $s_n^{(l-1)}$, the relation embedding $r_{nv}$, the relational adjacency matrix $M_{r_{nv}}$, the direction $D_{nv} \in \{h \to t, t \to h\}$, and the negation indicator $Neg_{nv} \in \{0, 1\}$. Then, we can calculate the message $m^{(l)}$ that $n$ passes to $v$:

$$m^{(l)} = \varrho(z_n^{(l-1)}, s_n^{(l-1)}, r_{nv}, M_{r_{nv}}, D_{nv}, Neg_{nv}). \tag{16}$$

The variable node $v$ receives $k_v$ messages from its neighbors, denoted as $m_1^{(l)}, \ldots, m_{k_v}^{(l)}$, where $k_v \geq 1$ represents the number of neighboring nodes in $N_{DP}(v)$. We employ product fuzzy logic to aggregate these messages and update the state (i.e., the symbolic vector) of node $v$:

$$s_v^{(l)} = \mathcal{N}(m_1^{(l)} \circ \cdots \circ m_{k_v}^{(l)}), \tag{17}$$

where $\circ$ is Hadmard product. While for $z_v^{(l)}$, we utilze the updated state $s_v^{(l)}$ to update it. Specifically, we form an entity set $\mathcal{V}_{nz}$ consisting of the entities corresponding to the non-zero elements in the fuzzy vector $s_v^{(l)}$. We then aggregate the embeddings of these entities weighted by their corresponding probabilities.

$$z_v^{(l)} = \sum_{i=1}^{|\mathcal{V}_{nz}|} s_{v,i}^{(l)} E_{e_i}, e_i \in \mathcal{V}_{nz}, \tag{18}$$

where $s_{v,i}^{(l)}$ is the corresponding probability of $e_i$ in $s_v^{(l)}$ and $E_{e_i}$ is the embedding of the entity $e_i$.

### 4.2.3 ANSWERING COMPLEX QUERIES WITH NSMP

According to Definition 5, a DNF query can be answered by solving all of its sub-conjunctive queries. For a given conjunctive query $Q$, we employ NSMP layers $L$ times to the query graph of $Q$, where $L$ is the depth of NSMP. Then, we use the state $s_y^{(L)}$ of the free variable node $y$ at the final layer, along with the corresponding embedding $z_y^{(L)}$, to obtain the probability of each entity being the answer, thereby retrieving the final answers:

$$p_{\mathcal{A}[\mathcal{Q}]} = \lambda s_y^{(L)} + (1 - \lambda) softmax(\underset{\forall e \in \mathcal{V}}{concat}(cos(z_y^{(L)}, E_e))), \tag{19}$$

where $\lambda$ is a hyperparameter that balances the influence of neural and symbolic representation and $cos(\cdot, \cdot)$ is the cosine similarity. Let $D$ denote the largest distance between the constant nodes and the free variable node. According to Wang et al. (2023c), $L$ should be larger than or equal to $D$ to ensure the free variable node successfully receives all messages from the constant nodes. Therefore, $L$ should dynamically change based on different types of conjunctive queries. In addition, the pre-trained neural link predictor is frozen in our work. In this case, NSMP has no trainable parameters.

### 4.3 DISCUSSION ON COMPUTATIONAL COMPLEXITY

As a neural-symbolic CQA model based on message passing, NSMP is more efficient than the current state-of-the-art step-by-step precise symbolic search method FIT (Yin et al., 2024). Next, we discuss the computational complexity of NSMP compared to FIT and reveal why message-passing-based models can lead to more efficient inference than the step-by-step approach. Here, we focus on time complexity, and the analyses of space complexity can be found in Appendix F.

Since the computational bottleneck of NSMP lies in the symbolic-related parts, we focus on discussing the time complexity of this aspect. Due to the sparsity of the adjacency matrix and fuzzy vector, both NSMP and FIT can utilize sparse techniques for efficient inference. But for simplicity, we will not consider this sparsity in the following discussion.

According to Equations 7 - 10, the complexity of symbolic one-hop inference is $\mathcal{O}(|\mathcal{V}|^2)$, so the complexity of neural-symbolic message encoding function $\varrho$ is approximately $\mathcal{O}(|\mathcal{V}|^2)$. This means that the complexity of message computation during message passing is linear to $\mathcal{O}(|\mathcal{V}|^2)$. For the node state update process, symbolic state update and neural state update are involved, corresponding to Equations 17 and 18, respectively. The complexity of the symbolic node state update is linear to $\mathcal{O}(|\mathcal{V}|)$, while the complexity of the neural node state update is $\mathcal{O}(|\mathcal{V}|d)$, where $d$ is the embedding dimension. Since we have $d \ll |\mathcal{V}|$, the total computational complexity of NSMP is approximately linear to $\mathcal{O}(|\mathcal{V}|^2)$. In particular, as a message-passing-based approach, the solving process of NSMP is the same for both acyclic and cyclic queries, i.e., performing message passing on the query graph. Consequently, the complexity of NSMP for any EFO$_1$ formula is approximately linear to $\mathcal{O}(|\mathcal{V}|^2)$.

For the previous step-by-step neural-symbolic method FIT, according to Yin et al. (2024), FIT solves the acyclic query by continuously removing constants and leaf nodes, and the complexity of this process is approximately linear to $\mathcal{O}(|\mathcal{V}|^2)$. However, for cyclic queries, FIT needs to enumerate one variable within the cycle as a constant node, so the complexity is $\mathcal{O}(|\mathcal{V}|^n)$, where $n$ is the number of variables in the query graph. In contrast, the complexity of NSMP on cyclic queries is approximately linear to $\mathcal{O}(|\mathcal{V}|^2)$, which means that NSMP can provide more efficient inference on cyclic queries. As we show in Section 5.2, NSMP achieves faster inference times when compared with FIT on cyclic queries, with speedup ranging from $69\times$ to over $150\times$. Moreover, while both NSMP and FIT exhibit a complexity linear to $\mathcal{O}(|\mathcal{V}|^2)$ for acyclic queries, the computation of different messages in NSMP's message-passing process is independent. This independence enables a parallelized computation of messages. In contrast, FIT requires removing constant and leaf nodes step by step, where the steps are interdependent, necessitating a serial process. As a result, even for acyclic queries, the message-passing-based NSMP can achieve more efficient inference. As demonstrated in Section 5.2, NSMP achieves a speedup of at least $10\times$ for acyclic queries in NELL995.

Table 1: MRR results of baselines and our model on BetaE datasets. The average score is calculated separately among positive and negative queries. Highlighted are the top **first** and **second** results.

| KG | Model | 1p | 2p | 3p | 2i | 3i | pi | ip | 2u | up | AVG.(P) | 2in | 3in | inp | pin | AVG.(N) |
|---|---|---|---|---|---|---|---|---|---|---|---|---|---|---|---|---|
| | BetaE | 39.0 | 10.9 | 10.0 | 28.8 | 42.5 | 22.4 | 12.6 | 12.4 | 9.7 | 20.9 | 5.1 | 7.9 | 7.4 | 3.6 | 6.0 |
| | CQD | 46.7 | 10.3 | 6.5 | 23.1 | 29.8 | 22.1 | 16.3 | 14.2 | 8.9 | 19.8 | 0.2 | 0.2 | 2.1 | 0.1 | 0.7 |
| | FuzzQE | 42.8 | 12.9 | 10.3 | 33.3 | 46.9 | 26.9 | 17.8 | 14.6 | 10.3 | 24.0 | 8.5 | 11.6 | 7.8 | 5.2 | 8.3 |
| | GNN-QE | 42.8 | 14.7 | 11.8 | 38.3 | 54.1 | 31.1 | 18.9 | 16.2 | 13.4 | 26.8 | 10.0 | 16.8 | 9.3 | 7.2 | 10.8 |
| | ENeSy | 44.7 | 11.7 | 8.6 | 34.8 | 50.4 | 27.6 | 19.7 | 14.2 | 8.4 | 24.5 | 10.1 | 10.4 | 7.6 | 6.1 | 8.6 |
| FB15k-237 | CQD$^A$ | 46.7 | 13.6 | 11.4 | 34.5 | 48.3 | 27.4 | 20.9 | 17.6 | 11.4 | 25.7 | 13.6 | 16.8 | 7.9 | 8.9 | 11.8 |
| | (Based on message passing) | | | | | | | | | | | | | | | |
| | LMPNN | 45.9 | 13.1 | 10.3 | 34.8 | 48.9 | 22.7 | 17.6 | 13.5 | 10.3 | 24.1 | 8.7 | 12.9 | 7.7 | 4.6 | 8.5 |
| | CLMPT | 45.7 | 13.7 | 11.3 | 37.4 | 52.0 | 28.2 | 19.0 | 14.3 | 11.1 | 25.9 | 7.7 | 13.7 | 8.0 | 5.0 | 8.6 |
| | NSMP | 46.7 | 15.1 | 12.3 | 38.7 | 52.2 | 31.2 | 23.3 | 17.2 | 11.9 | 27.6 | 11.9 | 17.6 | 10.8 | 7.9 | 12.0 |
| | BetaE | 53.0 | 13.0 | 11.4 | 37.6 | 47.5 | 24.1 | 14.3 | 12.2 | 8.5 | 24.6 | 5.1 | 7.8 | 10.0 | 3.1 | 6.5 |
| | CQD | 60.8 | 18.3 | 13.2 | 36.5 | 43.0 | 30.0 | 22.5 | 17.6 | 13.7 | 28.4 | 0.1 | 0.1 | 4.0 | 0.0 | 1.1 |
| | FuzzQE | 47.4 | 17.2 | 14.6 | 39.5 | 49.2 | 26.2 | 20.6 | 15.3 | 12.6 | 27.0 | 7.8 | 9.8 | 11.1 | 4.9 | 8.4 |
| | GNN-QE | 53.3 | 18.9 | 14.9 | 42.4 | 52.5 | 30.8 | 18.9 | 15.9 | 12.6 | 28.9 | 9.9 | 14.6 | 11.4 | 6.3 | 10.6 |
| | ENeSy | 59.0 | 18.0 | 14.0 | 39.6 | 49.8 | 29.8 | 24.8 | 16.4 | 13.1 | 29.4 | 11.3 | 8.5 | 11.6 | 8.6 | 10.0 |
| NELL995 | CQD$^A$ | 60.4 | 22.9 | 16.7 | 43.4 | 52.6 | 32.1 | 26.4 | 20.0 | 17.0 | 32.3 | 15.1 | 18.6 | 15.8 | 10.7 | 15.1 |
| | (Based on message passing) | | | | | | | | | | | | | | | |
| | LMPNN | 60.6 | 22.1 | 17.5 | 40.1 | 50.3 | 28.4 | 24.9 | 17.2 | 15.7 | 30.7 | 8.5 | 10.8 | 12.2 | 3.9 | 8.9 |
| | CLMPT | 58.9 | 22.1 | 18.4 | 41.8 | 51.9 | 28.8 | 24.4 | 18.6 | 16.2 | 31.3 | 6.6 | 8.1 | 11.8 | 3.8 | 7.6 |
| | NSMP | 60.7 | 21.6 | 17.5 | 44.2 | 53.8 | 33.7 | 26.7 | 19.1 | 14.4 | 32.4 | 12.4 | 15.7 | 13.7 | 7.8 | 12.4 |

Table 2: MRR results on FIT datasets. Highlighted are the top **first** and **second** results.

| Knowledge Graph | Model | pni | 2il | 3il | 2m | 2nm | 3mp | 3pm | im | 3c | 3cm | AVG |
|---|---|---|---|---|---|---|---|---|---|---|---|---|
| FB15k-237 | BetaE | 9.0 | 25.0 | 40.1 | 8.6 | 6.7 | 8.6 | 6.8 | 12.3 | 25.2 | 22.9 | 16.5 |
| | LogicE | 9.5 | 27.1 | 42.0 | 8.6 | 6.7 | 9.4 | 6.1 | 12.8 | 25.4 | 23.3 | 17.1 |
| | ConE | 10.8 | 27.6 | 43.9 | 9.6 | 7.0 | 9.3 | 7.3 | 14.0 | 28.2 | 24.9 | 18.3 |
| | QTO | 12.1 | 28.9 | 47.9 | 8.5 | 10.7 | 11.4 | 6.5 | 17.9 | 38.3 | 35.4 | 21.8 |
| | CQD | 7.7 | 29.6 | 46.1 | 6.0 | 1.7 | 6.8 | 3.3 | 12.3 | 25.9 | 23.8 | 16.3 |
| | FIT | 14.9 | 34.2 | 51.4 | 9.9 | 12.7 | 11.9 | 7.7 | 19.6 | 39.4 | 37.3 | 23.9 |
| | (Based on message passing) | | | | | | | | | | | |
| | LMPNN | 10.7 | 28.7 | 42.1 | 9.4 | 4.2 | 9.8 | 7.2 | 15.4 | 25.3 | 22.2 | 17.5 |
| | CLMPT | 10.1 | 31.0 | 48.5 | 8.7 | 7.8 | 10.1 | 6.1 | 15.8 | 30.2 | 28.5 | 19.7 |
| | NSMP | 13.4 | 32.9 | 51.2 | 9.2 | 9.9 | 11.4 | 7.5 | 18.9 | 39.0 | 34.5 | 22.8 |
| NELL995 | BetaE | 7.5 | 43.3 | 64.6 | 29.0 | 5.3 | 8.7 | 14.4 | 29.5 | 36.1 | 33.7 | 27.2 |
| | LogicE | 9.8 | 47.0 | 66.6 | 34.7 | 6.4 | 13.3 | 17.8 | 35.1 | 38.9 | 37.9 | 30.8 |
| | ConE | 10.3 | 42.1 | 65.8 | 32.4 | 7.0 | 12.6 | 16.8 | 34.4 | 40.2 | 38.2 | 30.0 |
| | QTO | 12.3 | 48.5 | 68.2 | 38.8 | 12.3 | 22.8 | 19.3 | 41.1 | 45.4 | 43.9 | 35.3 |
| | CQD | 7.9 | 48.7 | 68.0 | 31.7 | 1.5 | 12.9 | 13.8 | 33.9 | 38.8 | 35.9 | 29.3 |
| | FIT | 14.4 | 53.3 | 69.5 | 42.1 | 12.5 | 24.0 | 22.8 | 41.5 | 47.5 | 45.3 | 37.3 |
| | (Based on message passing) | | | | | | | | | | | |
| | LMPNN | 11.6 | 43.9 | 62.3 | 35.6 | 6.2 | 15.9 | 19.3 | 38.3 | 39.1 | 34.4 | 30.7 |
| | CLMPT | 12.5 | 48.7 | 68.2 | 36.6 | 7.5 | 19.0 | 19.9 | 39.1 | 44.4 | 41.2 | 33.7 |
| | NSMP | 13.0 | 52.4 | 71.3 | 37.6 | 11.5 | 21.7 | 18.3 | 41.7 | 46.6 | 42.4 | 35.7 |

## 5 EXPERIMENTS

### 5.1 EXPERIMENTAL SETTINGS

**Datasets and Queries.** We evaluate our model on two popular KGs: FB15k-237 (Toutanova & Chen, 2015) and NELL995 (Xiong et al., 2017). We follow Chen et al. (2022); Xu et al. (2022) and exclude FB15k (Bordes et al., 2013) since it suffers from major test leakage (Toutanova & Chen, 2015; Rossi et al., 2021). For a fair comparison with previous works, we evaluate our model using both the datasets introduced by Ren & Leskovec (2020), which we refer to as the **BetaE** datasets, and the datasets proposed by Yin et al. (2024), which we refer to as the **FIT** datasets. In particular, according to Yin et al. (2024), the "pni" query type in the BetaE datasets is not a real $EFO_1$ formula, we do not evaluate this query type in the BetaE datasets, but in the FIT datasets. For the graph representation of the query types and related statistics, please refer to Appendix B.

**Evaluation Protocol.** The evaluation scheme follows the previous works (Ren & Leskovec, 2020), which divide the answers to each complex query into easy and hard parts. The difference between easy and hard answers lies in whether they can be obtained through direct graph traversal. Such hard answers cannot be found directly from the observed knowledge graph, which means that the CQA model needs to complete non-trivial reasoning. Specifically, for each hard answer of a query, we rank it against non-answer entities and compute the Mean Reciprocal Rank (MRR).

**Baselines.** We consider the state-of-the-art CQA models from recent years as our baselines, including BetaE (Ren & Leskovec, 2020), CQD (Arakelyan et al., 2021), LogicE (Luus et al., 2021), ConE (Zhang et al., 2021), FuzzQE (Chen et al., 2022), GNN-QE (Zhu et al., 2022), ENeSy (Xu et al., 2022), CQD$^A$ (Arakelyan et al., 2023), LMPNN (Wang et al., 2023c), QTO (Bai et al., 2023b), CLMPT (Zhang et al., 2024a), and FIT (Yin et al., 2024), where LMPNN and CLMPT are based on message passing. We also compare more neural CQA models on the BetaE datasets in Appendix D. For details about the model, implementation and experiments, please refer to Appendix C.

### 5.2 MAJOR RESULTS

Table 1 and Table 2 present the results of NSMP compared to neural and neural-symbolic CQA baselines on the BetaE and FIT datasets, respectively. It can be observed that NSMP outperforms other message-passing-based CQA models on both positive and negative queries, even without training on complex queries, and achieves a significant improvement on negative queries. For other baselines, NSMP outperforms most neural and neural-symbolic CQA models, achieving a strong performance. Despite the achievement of second-best results compared to the state-of-the-art neural-symbolic model FIT, NSMP can offer superior inference efficiency. Specifically, we evaluate the relative speedup of NSMP over FIT in terms of inference time on the FIT dataset to make a sharp

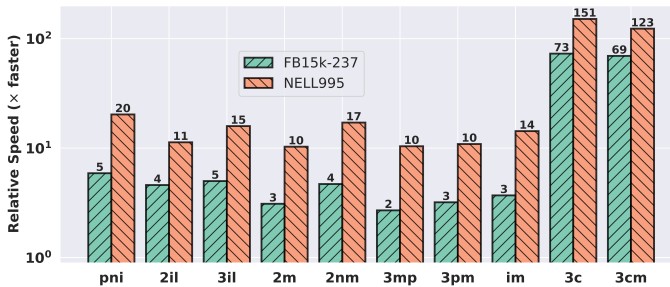

Figure 3: Relative speedup of NSMP over FIT in terms of inference time on FIT datasets.

contrast, as illustrated in Figure 3. NSMP demonstrates faster inference times across all query types on both FB15k-237 and NELL995, with speedup ranging from $2\times$ to over $150\times$. Notably, for more complicated cyclic queries, such as "3c" and "3cm", the relative speedup of NSMP becomes even more pronounced, as discussed in Section 4.3. Moreover, on the larger knowledge graph NELL995, NSMP exhibits even greater relative speedup, suggesting that NSMP offers better scalability compared to FIT. More experimental details on inference time can be found in Appendix C.

## 5.3 Ablation Study

We first explore the effect of the hyperparameter $L$ on performance. As noted in Section 4.2.3, $L$ should be greater than or equal to $D$. Thus, we evaluate the performance for depths $D, D+1, D+2$, and $D+3$ on FB15k-237. As shown in Table 3, NSMP achieves the best average performance when $L = D+1$. However, we observe that other depth choices can yield better results for specific query types. One approach is to manually select the most appropriate depth $L$ for different query types to achieve better average performance. As indicated in Table 3, this manual choice achieves an average MRR result of 23.3. But for simplicity, we set $L = D+1$ by default in our work.

To verify the effectiveness of the proposed dynamic pruning strategy, we conduct experiments on whether to perform dynamic pruning, and the results are shown in Table 4, where "w/o DP" indicates "without using dynamic pruning", AVG.(P), AVG.(N), and AVG.(F) represent the average scores for positive queries, negative queries on the BetaE datasets, and the average scores on the FIT datasets, respectively. It is found that the dynamic pruning strategy brings significant performance improvement, which shows the effectiveness of the strategy, indicating that the message from the variable node whose state is not updated is an unnecessary noise.

For the discussion of other hyperparameters, please refer to Appendix E. We also provide a case study in Appendix G.

| Model | pni | 2il | 3il | 2m | 2nm | 3mp | 3pm | im | 3c | 3cm | AVG |
|---|---|---|---|---|---|---|---|---|---|---|---|
| $L = D$ | 11.9 | 29.1 | 49.2 | **9.2** | 9.9 | 11.4 | **7.5** | **18.9** | **40.0** | **37.9** | 22.5 |
| $L = D+1$ | **13.4** | **32.9** | **51.2** | **9.2** | 9.9 | 11.4 | **7.5** | **18.9** | 39.1 | 34.5 | **22.8** |
| $L = D+2$ | 12.7 | **32.9** | **51.2** | 7.9 | **10.5** | **11.5** | 5.6 | 16.8 | 38.5 | 33.5 | 22.1 |
| $L = D+3$ | 13.3 | 31.8 | 50.2 | 7.9 | **10.5** | **11.5** | 5.6 | 16.8 | 38.1 | 31.8 | 21.8 |
| Manual Choice | **13.4** | **32.9** | **51.2** | **9.2** | **10.5** | **11.5** | **7.5** | **18.9** | **40.0** | **37.9** | **23.3** |

Table 3: MRR results of NSMP with different layers on FB15k-237.

| KG | Model | AVG.(P) | AVG.(N) | AVG.(F) |
|---|---|---|---|---|
| FB15k-237 | NSMP w/o DP | 27.0 | 11.6 | 20.4 |
| | NSMP | **27.6** | **12.0** | **22.8** |
| NELL995 | NSMP w/o DP | 32.1 | 12.1 | 31.2 |
| | NSMP | **32.4** | **12.4** | **35.7** |

Table 4: Average MRR results of NSMP with or without dynamic pruning.

## 6 Conclusion

In this paper, we propose NSMP, a neural-symbolic message passing framework, to answer complex queries over KGs. By integrating neural and symbolic reasoning, NSMP can utilize fuzzy logic theory to answer arbitrary $\text{EFO}_1$ queries without the need for training on complex query datasets, while also offering interpretability through fuzzy sets. Moreover, we introduce a novel dynamic pruning strategy to filter out unnecessary noisy messages during message passing. In our ablation study, we validate the effectiveness of this strategy. Extensive experimental results demonstrate that NSMP outperforms other message passing CQA models, achieving a strong performance, and can provide more efficient inference than previous state-of-the-art neural-symbolic CQA models.

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

## A  MORE RELATED WORKS

### A.1  NEURAL LINK PREDICTORS

Reasoning over KGs with missing knowledge is one of the long-standing topics in machine learning and has been widely explored. In the last few years, neural link predictors (Bordes et al., 2013; Yang et al., 2015; Trouillon et al., 2016; Dettmers et al., 2018; Sun et al., 2019; Chao et al., 2021) have been proposed to answer one-hop atomic queries on incomplete KGs. These latent feature models learn a low-dimensional vector for each entity and relation. By employing a well-defined scoring function to assess link confidence, they can effectively predict unseen links. In addition to these models, other research lines for one-hop KG reasoning include rule learning (Sadeghian et al., 2019; Cheng et al., 2022; 2023), text representation learning (Yao et al., 2019; Wang et al., 2021b; Saxena et al., 2022), reinforcement learning (Xiong et al., 2017; Das et al., 2018; Zhang et al., 2022), and graph neural networks (Schlichtkrull et al., 2018; Vashishth et al., 2020; Teru et al., 2020).

### A.2  NEURAL COMPLEX QUERY ANSWERING

Answering complex queries requires solving query formulas extended by atomic formulas combined with existential first-order logical operators. Therefore, CQA is a more challenging task than one-hop KG reasoning. In recent years, neural models have been proposed to represent the set of entities by low-dimensional embeddings. Most of them conceptualize a complex query as an operator tree (Ren et al., 2020; Wang et al., 2022a), wherein first-order logical operators are replaced with corresponding set operators. In particular, the existential quantifier induces a set projection operation, which corresponds to the logic skolemization (Luus et al., 2021). Specifically, these CQA models embed the sets of entities in various forms, including geometric shapes (Hamilton et al., 2018; Ren et al., 2020; Liu et al., 2021; Choudhary et al., 2021b; Zhang et al., 2021; Amayuelas et al., 2022; Bai et al., 2022; Nguyen et al., 2023), probabilistic distributions (Ren & Leskovec, 2020; Choudhary et al., 2021a; Yang et al., 2022; Long et al., 2022), fuzzy logic (Chen et al., 2022), and bounded histograms (Wang et al., 2023b). Subsequently, neural set operations are applied to these set embeddings in accordance with the operator tree to derive the answer embedding. Other neural

models represent the complex queries as graphs and solve CQA with graph neural networks (Daza & Cochez, 2020; Alivanistos et al., 2022) or transformers (Kotnis et al., 2021; Liu et al., 2022; Wang et al., 2023a; Bai et al., 2023a; Xu et al., 2023). However, most of the aforementioned neural models tend to be less effective than a simple neural link predictor on one-hop atomic queries.

To this end, Wang et al. (2023c) proposed a logical message passing model, which is most related to our work. This model leverages the pre-trained neural link predictor to infer intermediate embeddings for nodes in a query graph, interpreting these embeddings as logical messages. Through message passing paradigm (Xu et al., 2019), the free variable node embedding is updated to retrieve the answers. While effective on both one-hop atomic and multi-hop complex queries, logical message passing ignores the difference between constant and variable nodes, thus introducing noisy messages. To mitigate this, recent work (Zhang et al., 2024a) proposed a conditional message passing mechanism, which can be viewed as a pruning strategy regardless of the messages passed by variable nodes to constant nodes. However, this pruning strategy overlooks the noisy messages between variable nodes at the initial layers of message passing. In contrast, the dynamic pruning strategy employed in our model effectively eliminates these unnecessary noisy messages. Moreover, our proposed NSMP only needs to re-use a simple pre-trained neural link predictor without requiring training on complex query datasets, and offers interpretability through fuzzy sets.

### A.3 Symbolic Integration Models

In addition to the models mentioned above, there are several neural models enhanced with symbolic reasoning (Sun et al., 2020; Arakelyan et al., 2021; Zhu et al., 2022; Xu et al., 2022; Arakelyan et al., 2023; Bai et al., 2023b) related to our work. These neural-symbolic models typically integrate neural link predictors with fuzzy logic to address CQA. Most of them depend on the operator tree, which, as noted in Yin et al. (2024), can only handle the existential first order logic formulas in an approximate way. In contrast, our proposed neural-symbolic model, which utilizes the query graph, enables a more natural and direct handling of these formulas. A recently proposed neural-symbolic model (Yin et al., 2024) introduces an algorithm that cuts nodes and edges step by step to handle the query graph. Compared to this model, our message-passing-based approach can offer more efficient inference, as discussed in Section 4.3 and Section 5.2.

Table 5: Statistics of knowledge graphs as well as training, validation and test edge splits.

| Knowledge Graph | Entities | Relations | Training Edges | Val Edges | Test Edges | Total Edges |
|---|---|---|---|---|---|---|
| FB15k-237 | 14,505 | 237 | 272,115 | 17,526 | 20,438 | 310,079 |
| NELL995 | 63,361 | 200 | 114,213 | 14,324 | 14,267 | 142,804 |

Table 6: Statistics of different query types used in the BetaE datasets.

| Knowledge Graph | Training Queries | | Validation Queries | | Test Queries | |
|---|---|---|---|---|---|---|
| | 1p/2p/3p/2i/3i | 2in/3in/inp/pin/pni | 1p | Others | 1p | Others |
| FB15k-237 | 149,689 | 14,968 | 20,101 | 5,000 | 22,812 | 5,000 |
| NELL995 | 107,982 | 10,798 | 16,927 | 4,000 | 17,034 | 4,000 |

Table 7: Statistics of different query types used in the FIT datasets.

| Knowledge Graph | pni | 2il | 3il | 2m | 2nm | 3mp | 3pm | im | 3c | 3cm |
|---|---|---|---|---|---|---|---|---|---|---|
| FB15k-237 | 5,000 | 5,000 | 5,000 | 5,000 | 5,000 | 5,000 | 5,000 | 5,000 | 5,000 | 5,000 |
| NELL995 | 4,000 | 5,000 | 5,000 | 5,000 | 5,000 | 5,000 | 5,000 | 5,000 | 5,000 | 5,000 |

## B More Details about the Datasets

The statistics of the two knowledge graphs used in our experiment are shown in Table 5. As we described in Section 5.1, we evaluate our model using both BetaE (Ren & Leskovec, 2020) datasets

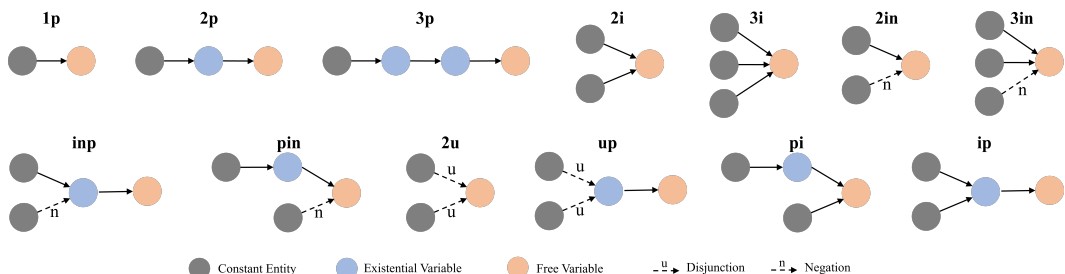

Figure 4: Graphical representation of the query types of the BetaE dataset considered in our experiment, where $p$, $i$, $u$, and $n$ represent projection, intersection, union, and negation, respectively.

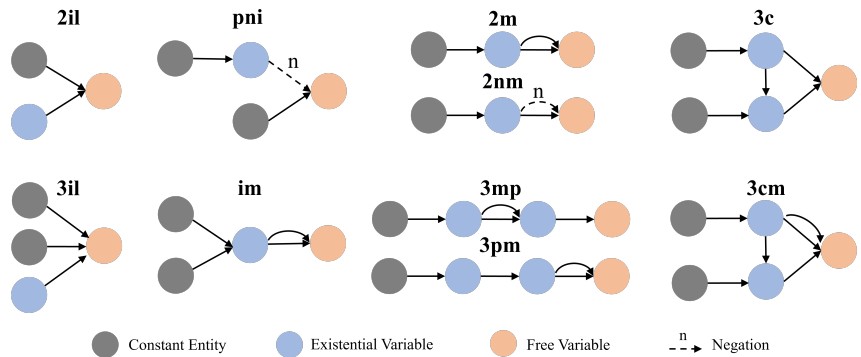

Figure 5: Graphical representation of the query types of the FIT dataset considered in our experiment, where $l$, $m$, and $c$ represent existential leaf, multi graph, and circle, respectively.

and FIT (Yin et al., 2024) datasets. The statistics for the BetaE datasets are shown in Table 6. Since our model does not require training on complex queries, we use only the test split of the BetaE datasets. In addition, according to Yin et al. (2024), the "pni" query type does not conform to a real $EFO_1$ formula. Consequently, we do not evaluate "pni" queries in BetaE datasets. Specifically, the query types we evaluated in BetaE datasets are shown in Figure 4. For the FIT dataset, which contains only test queries, the statistics are shown in Table 7. The query types in FIT datasets are shown in Figure 5.

## C DETAILS ABOUT THE MODEL, IMPLEMENTATION AND EXPERIMENTS

Our code is implemented using PyTorch. We use NVIDIA RTX 3090 GPU (24GB) and NVIDIA A100 GPU (40GB) to conduct all of our experiments. Following Arakelyan et al. (2021); Wang et al. (2023c); Yin et al. (2024), we select ComplEx-N3 (Trouillon et al., 2016; Lacroix et al., 2018) as the neural link predictor and use the checkpoints released by Arakelyan et al. (2021) for a fair comparison. The rank of ComplEx-N3 is 1,000, and the epoch for the checkpoints is 100. To determine the optimal hyperparameters of NSMP, we employ grid search. Specifically, the value for $\alpha$ is selected from $\{1, 10, 100, 1000, 10000, |\mathcal{V}|\}$, for $\lambda$ from $\{0, 0.1, 0.2, 0.3, 0.4, 0.5, 0.6, 0.7, 0.8, 0.9, 1.0\}$, and for $L$ from $\{D, D + 1, D + 2, D + 3\}$. After experimental analyses on hyperparameters, we choose $\lambda$ as 0.3 for FB15k-237 and 0.1 for NELL995. The number of layers $L$ is $D + 1$, and $\alpha$ is 100 for FB15k-237 and 1000 for NELL995.

In the case of choosing ComplEx-N3 as the neural link predictor, according to Wang et al. (2023c), given the corresponding complex embeddings, the neural message encoding function $\rho$ can be de-

rived into the following closed-form expressions for all four cases:

$$\rho(t, r, t \to h, 0) = \frac{\overline{r} \otimes t}{\sqrt{3\beta \|\overline{r} \otimes t\|}}, \tag{20}$$

$$\rho(h, r, h \to t, 0) = \frac{r \otimes h}{\sqrt{3\beta \|r \otimes h\|}}, \tag{21}$$

$$\rho(t, r, t \to h, 1) = \frac{-\overline{r} \otimes t}{\sqrt{3\beta \|\overline{r} \otimes t\|}}, \tag{22}$$

$$\rho(h, r, h \to t, 1) = \frac{-r \otimes h}{\sqrt{3\beta \|r \otimes h\|}}, \tag{23}$$

where $\beta$ is a hyperparameter that needs to be determined. In our application, we follow previous works (Wang et al., 2023c; Zhang et al., 2024a) and let $3\beta \|\cdot\| = 1$ for simplicity. Thus, $\rho$ is simplified to the following expressions for four cases:

$$\rho(t, r, t \to h, 0) := \overline{r} \otimes t, \tag{24}$$
$$\rho(h, r, h \to t, 0) := r \otimes h, \tag{25}$$
$$\rho(t, r, t \to h, 1) := -\overline{r} \otimes t, \tag{26}$$
$$\rho(h, r, h \to t, 1) := -r \otimes h. \tag{27}$$

For closed-form expressions of other neural link predictors, please refer to Wang et al. (2023c).

For the experiments evaluating the relative speedup of NSMP over FIT (Yin et al., 2024) in terms of inference time on the FIT dataset, we conduct the experiments on an NVIDIA A100 GPU. Specifically, we measure the average time required by NSMP and FIT to process each type of test query on the FIT dataset. To ensure a fair comparison, the batch size during the testing phase is set to 1. The relative speedup is presented in Figure 3, and detailed inference times are provided in Table 8.

Table 8: Inference time (ms/query) on each query type on FIT datasets, evaluated on one NVIDIA A100 GPU.

| Knowledge Graph | Model | pni | 2il | 3il | 2m | 2nm | 3mp | 3pm | im | 3c | 3cm |
|---|---|---|---|---|---|---|---|---|---|---|---|
| FB15k-237 | FIT | 93.4 | 48.4 | 73.4 | 47.8 | 71.2 | 69.4 | 74.4 | 69.8 | 2482.2 | 2824.2 |
| | NSMP | 15.8 | 10.6 | 14.6 | 15.2 | 15.2 | 25.6 | 23.4 | 18.8 | 34.0 | 40.6 |
| NELL995 | FIT | 843.0 | 419.0 | 630.6 | 422.6 | 649.6 | 636.0 | 632.4 | 634.2 | 11832.4 | 11149.9 |
| | NSMP | 41.5 | 37.0 | 39.6 | 41.0 | 38.0 | 61.2 | 57.8 | 44.2 | 78.2 | 90.4 |

## D    COMPARISON WITH MORE NEURAL CQA MODELS ON BETAE DATASETS

To further evaluate the performance of NSMP, we also consider comparing more neural CQA models on BetaE (Ren & Leskovec, 2020) datasets, including Q2P (Bai et al., 2022), MLP (Amayuelas et al., 2022), GammaE (Yang et al., 2022), CylE (Nguyen et al., 2023), WRFE (Wang et al., 2023b), and Pathformer (Zhang et al., 2024b). The reported MRR results are from these papers (Wang et al., 2023b; Nguyen et al., 2023; Zhang et al., 2024b). As shown in the results in Table 9, our model reaches the best performance across all query types on both FB15k-237 and NELL995, indicating the effectiveness of NSMP.

## E    ANALYSES ON MORE HYPERPARAMETERS

For the evaluation of hyperparameters $\alpha$ and $\lambda$, we conduct experiments on FB15k-237. Specifically, we compare the effects of different hyperparameters on model performance under default hyperparameter settings. Since $\alpha$ only influences the results of the negative queries, we evaluate $\alpha$ using the negative queries from both the BetaE and FIT datasets. As shown in the results in Table 10, $\alpha$ has a minor impact on the model performance, but $\alpha = 100$ achieves slightly better results, so we set $\alpha = 100$ for the experiments on FB15k-237. Similarly, we evaluate the performance of various settings for the hyperparameter $\lambda$ on the BetaE and FIT datasets. Specifically, we report the

Table 9: MRR results of other neural CQA models and our model on BetaE datasets. The average score is calculated separately among positive and negative queries. Highlighted are the top **first** results.

| Dataset | Model | 1p | 2p | 3p | 2i | 3i | pi | ip | 2u | up | AVG.(P) | 2in | 3in | inp | pin | AVG.(N) |
|---------|-------|----|----|----|----|----|----|----|----|----|---------|-----|-----|-----|-----|---------|
| | Q2P | 39.1 | 11.4 | 10.1 | 32.3 | 47.7 | 24.0 | 14.3 | 8.7 | 9.1 | 21.9 | 4.4 | 9.7 | 7.5 | 4.6 | 6.6 |
| | MLP | 42.7 | 12.4 | 10.6 | 31.7 | 43.9 | 24.2 | 14.9 | 13.7 | 9.7 | 22.6 | 6.6 | 10.7 | 8.1 | 4.7 | 7.5 |
| | GammaE | 43.2 | 13.2 | 11.0 | 33.5 | 47.9 | 27.2 | 15.9 | 13.9 | 10.3 | 24.0 | 6.7 | 9.4 | 8.6 | 4.8 | 7.4 |
| FB15k-237 | Pathformer | 44.8 | 12.9 | 10.6 | 34.2 | 47.3 | 26.2 | 17.0 | 14.9 | 10.0 | 24.2 | 6.4 | 11.6 | 8.3 | 4.7 | 7.8 |
| | CylE | 42.9 | 13.3 | 11.3 | 35.0 | 49.0 | 27.0 | 15.7 | 15.3 | 11.2 | 24.5 | 4.9 | 8.3 | 8.2 | 3.7 | 6.3 |
| | WRFE | 44.1 | 13.4 | 11.1 | 35.1 | 50.1 | 27.4 | 17.2 | 13.9 | 10.9 | 24.8 | 6.9 | 11.2 | 8.5 | 5.0 | 7.9 |
| | NSMP | **46.7** | **15.1** | **12.3** | **38.7** | **52.2** | **31.2** | **23.3** | **17.2** | **11.9** | **27.6** | **11.9** | **17.6** | **10.8** | **7.9** | **12.0** |
| | Q2P | 56.5 | 15.2 | 12.5 | 35.8 | 48.7 | 22.6 | 16.1 | 11.1 | 10.4 | 25.5 | 5.1 | 7.4 | 10.2 | 3.3 | 6.5 |
| | MLP | 55.2 | 16.8 | 14.9 | 36.4 | 48.0 | 22.7 | 18.2 | 14.7 | 11.3 | 26.5 | 5.1 | 8.0 | 10.0 | 3.6 | 6.7 |
| | GammaE | 55.1 | 17.3 | 14.2 | 41.9 | 51.1 | 26.9 | 18.3 | 15.1 | 11.2 | 27.9 | 6.3 | 8.7 | 11.4 | 4.0 | 7.6 |
| NELL995 | Pathformer | 56.4 | 17.4 | 14.9 | 39.9 | 50.4 | 26.0 | 19.4 | 14.4 | 11.1 | 27.8 | 5.1 | 8.6 | 10.3 | 3.9 | 7.0 |
| | CylE | 56.5 | 17.5 | 15.6 | 41.4 | 51.2 | 27.2 | 19.6 | 15.7 | 12.3 | 28.6 | 5.6 | 7.5 | 11.2 | 3.4 | 6.9 |
| | WRFE | 58.6 | 18.6 | 16.0 | 41.2 | 52.7 | 28.4 | 20.7 | 16.1 | 13.2 | 29.5 | 6.9 | 8.8 | 12.5 | 4.1 | 8.1 |
| | NSMP | **60.8** | **21.6** | **17.6** | **44.2** | **53.6** | **33.7** | **26.7** | **19.1** | **14.4** | **32.4** | **12.4** | **15.5** | **13.7** | **7.9** | **12.4** |

average MRR results for different $\lambda$ values on FB15k-237, as presented in Table 11, where AVG.(P), AVG.(N), and AVG.(F) represent the average scores for positive queries, negative queries on the BetaE datasets, and the average scores on the FIT datasets, respectively. Notably, except for $\lambda = 0$, the other hyperparameter settings exhibit comparable performance. Combined with the experimental results for hyperparameter $\alpha$, this indirectly validates the effectiveness of our proposed framework in maintaining stable results across different configurations. Furthermore, these results suggest that relying solely on aggregated neural representations (i.e., the $\lambda = 0$ case) is insufficient for addressing complex queries. We adopt a default setting of $\lambda = 0.3$, as it achieves a relatively balanced performance.

Table 10: MRR results for different hyperparameters $\alpha$ on FB15k-237.

| Model | 2in | 3in | inp | pin | pni | 2nm | AVG |
|-------|-----|-----|-----|-----|-----|-----|-----|
| $\alpha = 1$ | 11.9 | 17.6 | 10.7 | 8.0 | 13.0 | 8.9 | 11.7 |
| $\alpha = 10$ | 11.9 | 17.6 | 10.7 | 8.0 | 12.8 | 9.5 | 11.8 |
| $\alpha = 100$ | 11.9 | 17.6 | 10.8 | 7.9 | 13.4 | 9.9 | 11.9 |
| $\alpha = 1000$ | 11.9 | 17.6 | 10.8 | 7.9 | 13.7 | 9.6 | 11.9 |
| $\alpha = 10000$ | 11.7 | 17.6 | 10.8 | 7.8 | 13.9 | 9.0 | 11.8 |
| $\alpha = |\mathcal{V}|$ | 11.7 | 17.5 | 10.8 | 7.8 | 13.7 | 8.4 | 11.7 |

Table 11: Average MRR results for different hyperparameters $\lambda$ on FB15k-237.

| Model | AVG.(P) | AVG.(N) | AVG.(F) |
|-------|---------|---------|---------|
| $\lambda = 0$ | 19.8 | 5.5 | 17.4 |
| $\lambda = 0.1$ | 27.5 | 12.0 | 22.6 |
| $\lambda = 0.2$ | 27.6 | 12.0 | 22.7 |
| $\lambda = 0.3$ | 27.6 | 12.0 | 22.8 |
| $\lambda = 0.4$ | 27.6 | 12.0 | 22.8 |
| $\lambda = 0.5$ | 27.5 | 12.0 | 22.8 |
| $\lambda = 0.6$ | 27.5 | 12.0 | 22.8 |
| $\lambda = 0.7$ | 27.5 | 12.0 | 22.8 |
| $\lambda = 0.8$ | 27.4 | 12.0 | 22.8 |
| $\lambda = 0.9$ | 27.4 | 12.0 | 22.8 |
| $\lambda = 1.0$ | 27.4 | 12.0 | 22.9 |

## F  ANALYSES ON SPACE COMPLEXITY

For the space complexity of neural components, both NSMP and FIT use a pre-trained neural link predictor with a complexity of $\mathcal{O}((|\mathcal{V}|+|\mathcal{R}|)d)$, where $d$ is the embedding dimension. The symbolic

one-hop inference component of NSMP utilizes relational adjacency matrices, which contain $|\mathcal{R}| \cdot |\mathcal{V}|^2$ entries. However, due to the sparsity of KG, most entries are 0. With the help of sparse matrix techniques, the adjacency matrices can be stored efficiently. In this regard, NSMP has a space complexity similar to that of FIT. The neural adjacency matrix used in FIT also contains $|\mathcal{R}| \cdot |\mathcal{V}|^2$ entries and can be efficiently stored by setting appropriate thresholds.

## G  CASE STUDY

To verify whether NSMP can provide interpretability, we sample an "ip" query from FB15k-237 to visualize the corresponding entity ranking derived from the final fuzzy state of each variable after message passing. Specifically, the $\text{EFO}_1$ formula for the "ip" query we sampled is $\exists x, r_1(x, e_1) \wedge r_2(x, e_2) \wedge r_3(x, y)$, where $e_1$ is *Adventure Film*, $e_2$ is *The Expendables*, $r_1$ is *Genre*, $r_2$ is *Prequel*, and $r_3$ is *Film Regional Debut Venue*. This query has the following hard answers: *Paris*, *Buenos Aires*, *Madrid*, *Los Angeles*, *London* and *Belgrade*. After applying NSMP to the query, for each variable node, we find the top five entities with the highest probability from its final state based on Equation 19 and a softmax operation. The results are shown in Table 12. The results indicate that the fuzzy set of each variable can be used to represent its membership degrees across all entities, thereby providing interpretability. Although NSMP cannot sample a specific reasoning path like step-by-step methods Arakelyan et al. (2023); Bai et al. (2023b), such as the path sampled by beam search (Arakelyan et al., 2021), the introduction of fuzzy logic allows each variable's state to be represented as a fuzzy vector that defines a fuzzy set. These fuzzy vectors can be leveraged to enhance interpretability. Compared to neural embeddings as representations of variable states, fuzzy vectors offer a more intuitive explanation of the current state of variables, thereby improving interpretability.

Table 12: The top five entities with the highest probability for each variable and their corresponding probabilities. ✓ indicates that the correct entity is hit.

| $x$ | | $y$ | |
|---|---|---|---|
| Entity | Probability | Entity | Probability |
| *The Expendables 2* ✓ | 1.0 | *Buenos Aires* ✓ | 0.36 |
| *Green Lantern* | *almost* 0 | *Los Angeles* ✓ | 0.32 |
| *The Dark Knight* | *almost* 0 | *London* ✓ | 0.29 |
| *Battle Royale* | *almost* 0 | *Madrid* ✓ | 0.03 |
| *freak folk* | *almost* 0 | *Toronto International Film Festival* | *almost* 0 |

