# OpenReview forum: "Neural-Symbolic Message Passing with Dynamic Pruning"
_ICLR.cc/2025/Conference — Submitted to ICLR 2025_

### Official Review · Reviewer_d38M · 2024-10-22

**Soundness:** 1
**Presentation:** 1
**Contribution:** 1
**Rating:** 1
**Confidence:** 5

**Summary:**

The paper proposes a Neural-Symbolic Message Passing framework (NSMP) for complex logical query answering. NSMP uses the same components as many other baselines in the literature: a pre-trained link predictor (ComplEx-N3) as in CQD and FIT and the same message passing mechanism as in LMPNN. The only difference between NSMP and previous works is “dynamical pruning” which is a manual stop on message propagation from intermediate variables whose node states have not yet been updated. Experimentally, NSMP underperforms to GNN-QE and CQQ-A on BetaE datasets and underperforms to FIT on EFO-1 dataset while being somewhat faster than FIT.

**Strengths:**

N/A

**Weaknesses:**

The paper is either a direct plagiarism of [1] or a poor attempt to sell a very marginal change using the same template as [1] with a few changed words and butchered phrases.

Below are some examples of the significant text overlap with several replaced words:

* **Section 1: Introduction**

[1] `Knowledge graphs (KGs) store factual knowledge in the form of triples that can be utilized to support a variety of downstream tasks`

[this work] `Knowledge graphs (KGs) store factual knowledge in the form of graph representation, which can be applied to various intelligent application scenarios`

-----

[1] `However, given that modern KGs are usually auto-generated [48] or built through crowd-sourcing [52], so real-world KGs [9, 11, 45] are often considered noisy and incomplete, which is also known as the Open World Assumption [24, 28].`

[this work] `However, given that modern KGs are usually auto-generated () or built through crowd-sourcing (), real-world KGs () often suffer from incompleteness, which is also known as the Open World Assumption (OWA) ().` - note that even citations are the same.

-----

* **Section 2: Related Work** (almost verbatim content copy of the same sections and paragraphs), some particular examples:

[1] `Reasoning over KGs with missing knowledge is one of the fundamental problems in Artificial Intelligence and has been widely studied`

[this work] `Reasoning over KGs with missing knowledge is one of the long-standing topics in machine learning and has been widely explored`

-----

[1] `Other methods for link prediction include rule learning [42, 69], text representation learning [43, 53, 55], and GNNs [50, 70, 72].`

[this work] `... other research lines for one-hop KG reasoning include rule learning (), text representation learning (), reinforcement learning (), and graph neural networks ().` - again, references are the same

-----


[1] `… they embed entities and relations into continuous vector spaces and predict unseen triples by scoring triples with a well-defined scoring function. Such latent feature models can effectively answer one-hop atomic queries over incomplete KGs.`

[this work] `… neural link predictors () have been proposed to answer one-hop atomic queries on incomplete KGs. These latent feature models learn a low-dimensional vector for each entity and relation. By employing a well-defined scoring function to assess link confidence, they can effectively predict unseen links.`

-----

* **Section 2.2** is a rephrased version of the same paragraph from [1]

* **Section 3.1** (Knowledge Graphs) copies the content from Section 3 (Model-Theoretic Concepts for Knowledge Graphs) from [1]

* **Section 3.2** (Existential First Order Queries With a Single Free Variable) is almost a verbatim copy of Section 2.2 (EFO-1 Queries and Answers) from the FIT paper [2]

* **Section 3.3** (Query Graph) re-phrases the same section in [1]:

[1] `Since disjunctive queries can be solved in a scalable manner by transforming queries into the disjunctive normal form, it is only necessary to define query graphs for conjunctive queries`

[this work] `As mentioned above, since a DNF query can be addressed by solving all its sub-conjunctive queries, it is only necessary to define query graphs for conjunctive formulas`

-----

And copies the definitions 1-6  from [2]

* **Section 3.4** (Neural Link Predictors) copies the same from [1] and just replaces the model name CLMPT with NSMP
* **Section 3.5** (Neural One-Hop Inference on Atomic Formulas) re-uses the same-named subsection from [1]:

[1] `On each edge, when a node is at the head position, its neighbor is at the tail position, and vice versa.`

[this work] `On each edge, when a node is at the head position, its neighbor is at the tail position, and vice versa`

-----

[1] *Specifically, a logical message encoding function $\rho$ is proposed to perform one-hop inference.*

[this work] *Specifically, a logical message encoding function $\rho$ is proposed to define this neural one-hop inference on edges.*

Then, Equations 3-6 are exactly the same as Equations 6-9 from [1]

-----


* **Section 4** (Neural-Symbolic One-Hop Inference on Atomic Formulas) is largely based on ENeSy [3] (Section 4.2, page 8) without citing it: including the derivation of negation as $\frac{\alpha}{|\mathcal{V}|} - p_tM_r^{\top}$ and defining the encoding function as a softmax after concat where concat *is a function mapping the similarity between all entities $e \in \mathcal V$ and $\mathbf{v}$ to a vector* (ENeSy) and *is a function that maps the similarities between all entities and the intermediate embedding inferred by $\rho$ to a vector* (this work).

* **Section 5.1** (Message Passing with Dynamic Pruning) almost rephrases Section 4.1 (Conditional Logical Message Passing) from [1], they even start with the same sentence:

[1] `A query graph contains two types of nodes: constant entity nodes and variable nodes. `

[this work] `A query graph contains two types of nodes: constant and variable nodes`

-----

Then,

[1] `We decide whether to pass logical messages to a node based on its type.`

[this work] `we decide whether a variable node should pass messages to its neighboring variable nodes based on whether its state has been updated`


* **Section 5.2** (Node State Update Scheme) uses the same text and notation as Section 4.2 (Node Embedding Conditional Update Scheme) from [1] with small additions, eg,

[1] *Next, we discuss how to calculate the $z_e^a$ nd $z_v^l$ from the input layer $𝑙 = 0$ to latent layers $𝑙 > 0$*

[this work] *Next, we discuss how to compute these representations from the input layer $l = 0$ to latent layers $l > 0$*

-----

[1] For each neighbor node $n \in N(v)$, one can obtain information about the edge (i.e., the atomic formula) between $n$ and $v$, which contains the neighbor embedding $z_n^{(l −1)} \in D$, the relation $r_{nv} \in R$, the direction $D_{nv} \in \\{ℎ \rightarrow 𝑡, 𝑡 \rightarrow ℎ\\}$, and the negation indicator $Neg_{nv} \in \\{0, 1 \\}$

[4, LMPNN paper, Section 6.2] For each neighbor node $v \in N(n)$, one can obtain its embedding $z_v^{(l−1)} \in D$, the relation $r_{v \rightarrow n} \in R$, the direction $D_{v \rightarrow n} \in \\{h2t, t2h\\}$, and the negation indicator $Neg_{v \rightarrow n} \in \\{0, 1\\}$

[this work] For each neighboring node $n \in N_{DP} (v)$, one can obtain information about the edge between $n$ and $v$, which contains the neighbor mebdding $z_n^{(l-1)}$, the neighbor symbolic vector $s_n^{(l-1)}$, the relation embedding $r_{nv}$ , the relational adjacency matrix $M_{r_{nv}}$ , the direction $D_{nv} \in \\{h \rightarrow t, t \rightarrow h\\}$, and the negation indicator $Neg_{nv} \in \\{0, 1\\}$

It is obvious that [this work] just re-hashes the same content from [1] and [4] with a slightly different notation.

-----

Experimentally, the proposed NSMP underperforms well-established models on all benchmarks: worse than CQD-A on BetaE queries and worse than FIT on EFO-1 queries.

-----

Based on the above arguments, I do not see any scientific contribution in this manuscript, the issue has to be elevated to ACs, SACs, and PCs. It is sad that in this submission we have to measure the degree of plagiarism and deception instead of novel scientific contributions.


[1] Zhang et al. Conditional Logical Message Passing Transformer for Complex Query Answering. KDD 2024.
[2] Yin et al. Rethinking Complex Queries on Knowledge Graphs with Neural Link Predictors. ICLR 2024.
[3] Xu et al. Neural-Symbolic Entangled Framework for Complex Query Answering. NeurIPS 2022.
[4] Wang et al. Logical Message Passing Networks with One-hop Inference on Atomic Formulas. ICLR 2023.

**Questions:**

N/A

**Details Of Ethics Concerns:**

Significant text overlap (up to the same exact sentences and formulations) as Zhang et al. Conditional Logical Message Passing Transformer for Complex Query Answering (KDD 2024).

---

> ### Author Response · Authors · 2024-11-21
> **(1/4)**
>
> Thank you for your comments, and below I will respond point by point to potential misunderstandings and your allegations of plagiarism. Looking forward to your reply.
>
> **For Summary:**
>
> > NSMP uses the same components as many other baselines in the literature: a pre-trained link predictor (ComplEx-N3) as in CQD and FIT and the same message passing mechanism as in LMPNN.
>
> * First, pre-trained neural link predictors have been widely utilized in various studies, including previous message passing CQA models [1,2] based on pre-trained link predictors. As a message passing CQA model, NSMP's main contribution does not lie in the introduction of pre-trained link predictors but in integrating the neural reasoning capabilities of pre-trained link predictors with the symbolic reasoning capabilities of TensorLog [3]. Leveraging fuzzy logic, this approach enables message passing CQA methods to generalize to arbitrary EFO1 queries without requiring training on complex queries while providing interpretable answers, akin to other step-by-step neural-symbolic methods [4,5,6].
> * Second, we do not adopt the same message passing mechanism as LMPNN [2]. Our proposed framework comprises both neural and symbolic components, and we explicitly state in the paper that we use the neural message encoding function from LMPNN as the neural one-hop inference module in our framework. Beyond this, the symbolic one-hop inference module and the neural-symbolic one-hop inference module in our framework are not related to LMPNN. This means that NSMP is quite different from LMPNN in terms of message computation. Furthermore, the message passing mechanism and node state update scheme in our proposed NSMP differ significantly from those in LMPNN. Specifically, we compute neural-symbolic messages based on a novel dynamic pruning strategy, rather than the bidirectional unconditional message passing used in LMPNN or the conditional message passing employed in CLMPT [1]. Additionally, we update node states based on fuzzy logic rather than the complex neural networks used in LMPNN and CLMPT.
>
> > The only difference between NSMP and previous works is "dynamical pruning" which is a manual stop on message propagation from intermediate variables whose node states have not yet been updated.
>
> Dynamic pruning is not a manual operation; instead, it adapts automatically and dynamically based on different query graphs, with the pruning process varying across different query graphs. Dynamic pruning is one of the contributions of our paper. However, it is not the only difference from previous works as you said. To address potential misunderstandings, we believe it is necessary to clarify our contributions. In summary, the contributions of our paper are as follows:
> * We propose a neural-symbolic message passing framework that, for the first time, integrates neural and symbolic reasoning within a message passing CQA model. By leveraging fuzzy logic theory, our approach can naturally answer arbitrary existential first order logic formulas without requiring training on complex queries while providing interpretability.
> * We propose a novel dynamic pruning strategy to filter out unnecessary noisy messages between variable nodes during message passing, thereby reducing unnecessary computation and improving performance.
> * Extensive experimental results on benchmark datasets show that NSMP achieves a strong performance. In particular, NSMP significantly improves the performance of message passing CQA models on negative queries by introducing symbolic reasoning and fuzzy logic.
> * Through computational complexity analyses, we reveal that our message-passing-based method can provide more efficient inference than the current state-of-the-art step-by-step neural-symbolic method, and empirically verify this.
>
> We uploaded a revised PDF where we clarify the contributions mentioned above in the Introduction section.
>
> Due to the space limitation, please see the following comment, i.e., (2/4).

---

> > ### Author Response · Authors · 2024-11-21
> > **(2/4)**
> >
> > > Experimentally, NSMP underperforms to GNN-QE and CQQ-A on BetaE datasets and underperforms to FIT on EFO-1 dataset while being somewhat faster than FIT.
> >
> >
> > We believe there may be some misunderstandings regarding the evaluation of the experimental results.
> > * First, NSMP demonstrates consistently superior average performance over GNN-QE [7] on the BetaE datasets and overall outperforms $\text{CQD}^A$ [8]. The average MRR results of these models on FB15k-237 are as follows, as reported in Table 1 of our paper:
> >     | Model  | AVG.(P) | AVG.(N) |
> >     | ------ | ------- | ------- |
> >     | GNN-QE | 26.8    | 10.8    |
> >     | $\text{CQD}^A$  | 25.7    | 11.8    |
> >     | NSMP   | **27.6**    | **12.0**    |
> >
> >   The average MRR results of these models on NELL995 are as follows, as reported in Table 1 of our paper:
> >     | Model  | AVG.(P) | AVG.(N) |
> >     | ------ | ------- | ------- |
> >     | GNN-QE | 28.9    | 10.6    |
> >     | $\text{CQD}^A$  | 32.3    | **15.1**    |
> >     | NSMP   | **32.4**    | 12.4    |
> >
> >   From the results above, NSMP outperforms $\text{CQD}^A$ and GNN-QE in all cases except for negative queries on NELL995, where its performance is lower than that of $\text{CQD}^A$. However, $\text{CQD}^A$ requires training on complex queries, whereas NSMP does not. Therefore, overall, NSMP can be considered superior to $\text{CQD}^A$.
> > * Second, although NSMP achieves second-best results compared to the state-of-the-art neural-symbolic method FIT [6], it offers significantly more efficient inference. In Section 4.3 of the revised paper, we discuss the computational complexity of NSMP and FIT, revealing that the message-passing-based NSMP can provide more efficient inference than the step-by-step approach of FIT and empirically verifying this. Refer to Figure 3 and Table 8 of the paper: on the FIT datasets, NSMP achieves up to over 150$\times$ relative speedup in inference time compared to FIT. Furthermore, on the larger knowledge graph NELL995, NSMP achieves over 10$\times$ speedup across various query types. This stands in contrast to the characterization of NSMP as ```"being somewhat faster than FIT"```.
> >
> >
> > **For the Allegations of Plagiarism:**
> >
> > > The paper is either a direct plagiarism of [1] or a poor attempt to sell a very marginal change using the same template as [1] with a few changed words and butchered phrases.
> >
> > We believe the contributions of our paper have been clearly clarified in the above discussion. Below, we will address your allegations of "plagiarism" point by point. It is important to note upfront that the examples of text overlap you have cited are entirely unrelated to the method and contributions proposed in our work.
> >
> > **For the Allegations of plagiarism in Section1 and Section2:**
> >
> > As a paper on knowledge graph reasoning and complex query answering, we consider it entirely reasonable for there to be similarities in the descriptions of the research background and recent advancements with previous works in the same domain. It is important to emphasize that these similar descriptions have no bearing on the method and contributions proposed in this paper. In fact, in both the Introduction and Related Work sections, we provide a detailed discussion of the limitations of previous methods and the solutions we propose. We thoroughly analyze the distinctions between NSMP and previous message passing CQA approaches, highlight our contributions, and discuss other neural-symbolic CQA methods.
> >
> > Due to the space limitation, please see the following comment, i.e., (3/4).

---

> > > ### Author Response · Authors · 2024-11-21
> > > **(3/4)**
> > >
> > > **For the Allegations of plagiarism in Section3:**
> > >
> > >
> > > The Background section covers the necessary background knowledge, definitions, and methods or concepts proposed in previous works that are essential for understanding this paper. As mentioned above, as a message passing CQA method designed for complex query answering over knowledge graphs, it is natural for the background descriptions to resemble those in previous message passing CQA methods. Specifically, to accurately define the KG-related concepts, the $\text{EFO}_1$ formula, and the query graph, we referenced the definitions provided in [1,2,6,9]. We have explicitly stated in the paper that we follow previous works in adopting these definitions. As part of the background, these definitions are unrelated to the contributions of this paper. For Section 3.4 (Neural Link Predictors) and Section 3.5 (Neural One-Hop Inference on Atomic Formulas), NSMP utilizes the neural one-hop inference function based on pre-trained neural link predictors, as employed in LMPNN and CLMPT. Hence, we included descriptions of these components in the background section. Similar to CLMPT, NSMP directly adopts the neural message encoding function from LMPNN as the neural component of its message passing framework. Therefore, it is natural for these descriptions to resemble those in LMPNN and CLMPT. Importantly, these descriptions are unrelated to the core contributions of this paper. The core contribution of this paper lies in introducing symbolic reasoning and fuzzy logic into the neural logic message passing mechanism of previous works. The content related to neural one-hop inference is only introduced as background to explain the proposed method of this paper.
> > >
> > > **For the Allegations of plagiarism in Section4:**
> > >
> > > * First, regarding the example you raised concerning Equation 10 (i.e., $\frac{\alpha}{|\mathcal{V}|} - p_t M_{r}^{\top}$), this equation is directly derived from the standard fuzzy logic negator $n_{stand}(p) = 1 - p$. We have clearly stated in the paper that this equation is defined based on fuzzy logic theory and have cited the related works. While we referenced a trick from ENeSy [5] that incorporates a hyperparameter to handle this fuzzy logic negator, the equation fundamentally originates from the standard fuzzy logic negator $n_{stand}(p) = 1 - p$. In the revised paper, we have added a citation to ENeSy.
> > > * Second, regarding the example you raised concerning Equation 11 (i.e., $f(\rho) = \text{softmax}\left(\underset{\forall e \in \mathcal{V}}{\text{concat}}\left(\mathcal{S}(\rho, E_{e})\right)\right)$), this is essentially a standard softmax operation. For greater clarity, we referred to the equation notation in ENeSy, another neural-symbolic method. However, this is merely a representation format for the equation, and adopting a particular formulation does not imply the use of another method.
> > > * Third, we cited ENeSy in the original paper and used it as a baseline for comparison. It is worth mentioning that from the MRR results in Table 1, the performance of our proposed NSMP is significantly better than ENeSy.
> > >
> > > Due to the space limitation, please see the following comment, i.e., (4/4).

---

> > > > ### Author Response · Authors · 2024-11-21
> > > > **(4/4)**
> > > >
> > > > **For the Allegations of plagiarism in Section5:**
> > > >
> > > > As a message-passing-based CQA method, it is reasonable for NSMP to share similarities in description with other message-passing-based CQA methods (i.e., LMPNN and CLMPT), as these methods all involve message passing processes and node state update mechanisms. However, this does not imply that NSMP operates identically to previous message passing CQA models. Notably, NSMP differs from previous message passing methods in three fundamental aspects, which underpin its main contributions. These distinctions are: (1) symbolically integrated message computation (Section 4.1 in the revised paper), (2) a non-parametric node state update scheme based on fuzzy logic (Section 4.2.2 in the revised paper), and (3) a dynamic pruning strategy (Section 4.2.1 in the revised paper). Therefore, regarding the claim of ```"a poor attempt to sell a very marginal change"```, we believe there is a misunderstanding. In fact, to clearly and directly highlight the distinctions between NSMP and previous message passing CQA methods, we deliberately referenced the descriptions of message-passing in previous works to better articulate the essential differences between NSMP and these previous approaches.
> > > >
> > > >
> > > >
> > > >
> > > > **For Misunderstandings about Experimental Results:**
> > > >
> > > > > Experimentally, the proposed NSMP underperforms well-established models on all benchmarks: worse than CQD-A on BetaE queries and worse than FIT on EFO-1 queries.
> > > >
> > > > As discussed above, NSMP outperforms $\text{CQD}^A$ without requiring training on complex queries. Compared to FIT, NSMP achieves competitive performance while offering significantly more efficient inference, with relative inference time speedups of up to over 150$\times$.
> > > >
> > > >
> > > >
> > > > [1] Conditional logical message passing transformer for complex query answering. KDD 2024
> > > >
> > > > [2] Logical message passing networks with one-hop inference on atomic formulas. ICLR 2023
> > > >
> > > > [3] Tensorlog: A probabilistic database implemented using deep-learning infrastructure. JAIR 2020
> > > >
> > > > [4] Complex query answering with neural link predictors. ICLR 2021
> > > >
> > > > [5] Neural-symbolic entangled framework for complex query answering. NeurIPS 2022
> > > >
> > > > [6] Rethinking complex queries on knowledge graphs with neural link predictors. ICLR 2024
> > > >
> > > > [7] Neural-Symbolic Models for Logical Queries on Knowledge Graphs. ICML 2022
> > > >
> > > > [8] Adapting neural link predictors for data-efficient complex query answering. NeurIPS 2023
> > > >
> > > > [9] Logical queries on knowledge graphs: Emerging interface of incomplete relational data. IEEE Data Eng. Bull. 2022

---

> > > > > ### Comment · Reviewer_d38M · 2024-11-21
> > > > >
> > > > > > we consider it entirely reasonable for there to be similarities in the descriptions of the research background and recent advancements with previous works in the same domain
> > > > >
> > > > > > it is natural for the background descriptions to resemble those in previous message passing CQA methods
> > > > >
> > > > > The authors confirm that the copy-pasting was indeed intentional. It is absolutely unacceptable to frame verbatim word-by-word copy-pasting from several related works as *“some similarities and resemblances”* and presenting the rest as an outstanding contribution. I am strongly convinced this is not the level of scientific excellence ICLR is looking for and stand for a strong reject with flagging the paper to ACs and PCs.

---

### Official Review · Reviewer_fVNG · 2024-10-31

**Soundness:** 2
**Presentation:** 4
**Contribution:** 1
**Rating:** 3
**Confidence:** 4

**Summary:**

This paper proposes Neural-Symbolic Message Passing (NSMP) to integrates neural and symbolic reasoning for answering complex queries on knowledge graphs. It adopts pre-trained neural link predictors and fuzzy logic operations to estimate the embeddings and the fuzzy set for each variable in a given query respectively. NSMP doesn’t require to be trained on complex query datasets, and is claimed to provide interpretable answers. The authors further introduce a dynamic pruning strategy to filter out noisy messages in the message passing process. Empirically, NSMP achieves competitive performance on both BetaE and FIT datasets, even though most baselines are trained on complex queries.

**Strengths:**

- NSMP achieves competitive performance against state-of-the-art methods on complex queries, even though NSMP doesn’t require to be trained on complex queries. The results should be deemed as solid.
- The writing of this paper is generally good except for the abstract and the intro. It’s easy to comprehend most technical details of this paper.

**Weaknesses:**

- The contributions of this paper are not clear. The authors claimed three challenges in the intro: bad performance on negative queries, noisy messages and interpretability, but there is no clear correspondence between the model components and the challenges.
- Most components of NSMP have limited novelty regarding the literature of complex queries. One-hop inference and message passing (Section 3.5 & 5) is almost identical to LMPNN[1]. Combining neural and symbolic representations (Section 4 & 5) is very close to EmQL[2], which the authors didn’t even cite in the paper. The idea of using non-parameteric fuzzy logic to avoid training shares many spirits with CQD[3], though the actual design is different.
- The usage of fuzzy logic in this paper is not standard compared to [3, 4, 5]. In [3, 4, 5], every entity has a probability to be bound to a variable, independent of other entities. This enables fuzzy logic operations to form a closure and satisfy logic laws (e.g. De Morgan’s laws). In this paper, all entities form a distribution to be bound to a variable and have to be normalized after every operation (Equation 7-15, 17). As a result, NSMP doesn’t satisfy De Morgan’s laws and has to resort to DNF to handle disjunctions.
- The authors claimed interpretability as an advantage of NSMP, but there are no supporting qualitative experiments. I would expect some experiments to visualize the intermediate variables in complex queries similar to [3, 4].

[1] Logical Message Passing Networks with One-hop Inference on Atomic Formulas. Wang et al. ICLR 2023.

[2] Faithful Embeddings for Knowledge Base Queries. Sun et al. NeurIPS 2020.

[3] Complex Query Answering with Neural Link Predictors. Arakelyan, Daza and Minervini et al. ICLR 2021.

[4] Neural-Symbolic Models for Logical Queries on Knowledge Graphs. Zhu et al. ICML 2022.

[5] Rethinking Complex Queries on Knowledge Graphs with Neural Link Predictors. Yin et al. ICLR 2024.

**Questions:**

- In my opinion, dynamic pruning refers to pruning strategies that depend on the learned representations. Here the pruning strategy only relies on the query graph and the layer index, which should be regarded as a static pruning strategy. A better alternative is to call it Bellman-Ford iteration or breath-first search. You may refer to [6, 7] for more insights.
- Line 473-475: It looks like that the hyperparameters of NSMP have to be tuned for each dataset. Could you provide an analysis on hyperparameter sensitivity? Do there exist some robust default hyperparameters?
- Line 20-23: The logic of this sentence is weird. How does “re-using a simple pre-trained neural link predictors” serve as an approach to “generalizes to complex queries based on fuzzy logic theory”?
- Line 132-133: What do you mean by “the more complex existential first order logic formulas”? Is it a disadvantage?
- Line 182: Is DNF really scalable for handling disjunction operators? I feel the opposite.
- Equation 9 & 10: What if we get negative values after subtraction?
- Typos:
    - Line 30: graph representation → graph representations
    - Line 34: A more straightforward → A straightforward
    - Line 261: symbolic representation → symbolic representations

[6] Neural Bellman-Ford Networks: A General Graph Neural Network Framework for Link Prediction. Zhu et al. NeurIPS 2021.

[7] Distance-Based Propagation for Efficient Knowledge Graph Reasoning. Shomer et al. EMNLP 2023.

---

> ### Author Response · Authors · 2024-11-21
> **(1/3)**
>
> Thanks for your valuable comments. We hope the following response addresses your concern.
>
> **For W1 and W2:**
>
> In light of the potential for misunderstanding here, we would like to clarify our contribution:
> * We propose a neural-symbolic message passing framework that, for the first time, integrates neural and symbolic reasoning within a message passing CQA model. By leveraging fuzzy logic theory, our approach can naturally answer arbitrary existential first order logic formulas without requiring training on complex queries while providing interpretability.
> * We propose a novel dynamic pruning strategy to filter out unnecessary noisy messages between variable nodes during message passing, thereby reducing unnecessary computation and improving performance.
> * Extensive experimental results on benchmark datasets show that NSMP achieves a strong performance. In particular, NSMP significantly improves the performance of message passing CQA models on negative queries by introducing symbolic reasoning and fuzzy logic.
> * Through computational complexity analyses, we reveal that our message-passing-based method can provide more efficient inference than the current state-of-the-art step-by-step neural-symbolic method, and empirically verify this.
>
> We uploaded a revised PDF where we clarify the contributions mentioned above in the Introduction section.
>
> > The authors claimed three challenges in the intro: bad performance on negative queries, noisy messages and interpretability, but there is no clear correspondence between the model components and the challenges.
>
> With the introduction of fuzzy sets, we can naturally leverage fuzzy logic to handle the negation operator, significantly improving the performance of message-passing-based CQA methods on negative queries. Furthermore, the incorporation of fuzzy sets enables NSMP to provide interpretable answers. To address the challenge of noisy messages between variables, we propose a novel dynamic pruning strategy.
>
> > One-hop inference and message passing (Section 3.5 & 5) is almost identical to LMPNN[1].
>
> We do not adopt the same message passing mechanism as LMPNN [1]. Our proposed framework comprises both neural and symbolic components, and we explicitly state in the paper that we use the neural message encoding function from LMPNN as the neural one-hop inference module in our framework. Beyond this, the symbolic one-hop inference module and the neural-symbolic one-hop inference module in our framework are not related to LMPNN. This means that NSMP is quite different from LMPNN in terms of message computation. Furthermore, the message passing mechanism and node state update scheme in our proposed NSMP differ significantly from those in LMPNN. Specifically, we compute neural-symbolic messages based on a novel dynamic pruning strategy, rather than the bidirectional unconditional message passing used in LMPNN or the conditional message passing employed in CLMPT [2]. Additionally, we update node states based on fuzzy logic rather than the complex neural networks used in LMPNN and CLMPT.
>
> > Combining neural and symbolic representations (Section 4 & 5) is very close to EmQL[2], which the authors didn’t even cite in the paper.
>
> Our proposed framework is fundamentally different from EmQL [3]. Specifically, our core contribution does not lie in combining neural and symbolic representations but in incorporating symbolic reasoning and fuzzy logic into the logical message passing mechanism [1]. This integration enables message-passing-based CQA methods to generalize to arbitrary $\text{EFO}_1$​ queries based on pre-trained link predictors, without requiring additional training, similar to other step-by-step methods [4,5,6,8]. Furthermore, our approach leverages the advantages of the message passing mechanism to provide more efficient inference compared to step-by-step methods (see Section 4.3 of the revised paper). Additionally, we have added a citation to EmQL in the related work section of the revised manuscript.
>
> > The idea of using non-parameteric fuzzy logic to avoid training shares many spirits with CQD[3], though the actual design is different.
>
> CQD [4] is a pioneering work in CQA, and many subsequent studies [5,6,7,8,9] have drawn inspiration from CQD's use of non-parametric fuzzy logic. As a neural-symbolic CQA method, it is natural that NSMP shares many spirits with CQD in this regard.
>
> Due to the space limitation, please see the following comment, i.e., (2/3).

---

> > ### Author Response · Authors · 2024-11-21
> > **(2/3)**
> >
> > **For W3:**
> >
> > After normalization operation, each intermediate state becomes a fuzzy vector $p \in [0,1]^{1\times |\mathcal{V}|}$, enabling direct application of fuzzy logic operations, such as conjunction and disjunction, on these states. The normalization operation in Equation 17 is applied just because our implementation includes a step to clip small values based on a predefined threshold. In principle, NSMP could handle disjunctive queries using the fuzzy logic disjunction operation, such as $x+y-x\cdot y$, rather than relying on the DNF-based approach. However, the current definition of query graph does not support this operation. We previously experimented with manually designing the fuzzy logic disjunction operation for specific query types (e.g., "2u" and "up"), but this approach required separate handling for different types of disjunctive queries, limiting its flexibility. Moreover, our preliminary experimental results showed that this approach underperformed compared to the DNF-based approach. Therefore, we followed prior work and adopted DNF as the default setting.
> >
> >
> > **For W4:**
> >
> > Compared with previous message passing CQA methods, NSMP can provide interpretable answers. In the revised paper, we have included a case study (see Appendix G) analyzing the interpretability of NSMP, focusing on the fuzzy sets associated with different variables. The fuzzy set of each variable can be used to represent its membership degrees across all entities, thereby providing interpretability. Although NSMP cannot sample a specific reasoning path like step-by-step methods [6,7], such as the path sampled by beam search [4], the introduction of fuzzy logic allows each variable's state to be represented as a fuzzy vector that defines a fuzzy set. These fuzzy vectors can be leveraged to enhance interpretability. Compared to neural embeddings as representations of variable states, fuzzy vectors offer a more intuitive explanation of the current state of variables, thereby improving interpretability.
> >
> >
> > **For Q1:**
> >
> > > Here the pruning strategy only relies on the query graph and the layer index, which should be regarded as a static pruning strategy.
> >
> > The "dynamic" nature of our proposed dynamic pruning lies in the fact that the strategy adapts the pruning process dynamically for different query graphs. In our implementation, we do not design pruning strategies specifically based on the query graph and layer index. Instead, we dynamically filter out noisy messages between variable nodes based on whether the state of the variable node has been updated. Therefore, we refer to this approach as dynamic pruning.
> >
> > **For Q2:**
> >
> > > It looks like that the hyperparameters of NSMP have to be tuned for each dataset. Could you provide an analysis on hyperparameter sensitivity? Do there exist some robust default hyperparameters?
> >
> > Based on the analyses in Section 5.3 (Ablation Study) and Appendix E (Analyses on More Hyperparameters) of our revised paper, we conducted experiments on FB15k-237 to investigate the impact of various hyperparameters. According to the analyses presented in the paper, NSMP demonstrates stable performance across different hyperparameter settings (see Table 10 and Table 11), indicating its robustness to hyperparameter variations. Even when applying the hyperparameter configuration optimized for FB15k-237 (i.e., $\lambda=0.3$, $\alpha=100$) to NELL995, NSMP maintains stable performance on NELL995. The results are shown in the table below:
> > | Model              | AVG.(P) | AVG.(N) | AVG.(F) |
> > | ------------------ | ------- | ------- | ------- |
> > | $\lambda=0.3$, $\alpha=100$     | 32.3    | 12.4    | 35.6    |
> > | $\lambda=0.1$, $\alpha=1000$ (Best Choice) | 32.4    | 12.4    | 35.7    |
> >
> > These results suggest that NSMP is insensitive to hyperparameter choices and offers a robust default hyperparameter configuration.
> >
> > Due to the space limitation, please see the following comment, i.e., (3/3).

---

> > > ### Author Response · Authors · 2024-11-21
> > > **(3/3)**
> > >
> > > **For Q3:**
> > >
> > > > Line 20-23: The logic of this sentence is weird. How does “re-using a simple pre-trained neural link predictors” serve as an approach to “generalizes to complex queries based on fuzzy logic theory”?
> > >
> > > Apologies for any confusion caused by this sentence. In the revised version of the paper, we will replace it with the following clearer description: "In this paper, we propose a Neural-Symbolic Message Passing framework (NSMP) based on pre-trained neural link predictors. By introducing symbolic reasoning and fuzzy logic, NSMP can generalize to arbitrary existential first order logic queries without requiring training on any complex queries while providing interpretable answers."
> > >
> > >
> > > **For Q4:**
> > >
> > > > Line 132-133: What do you mean by “the more complex existential first order logic formulas”? Is it a disadvantage?
> > >
> > > Apologies for the misunderstanding caused by this sentence. In the revised version of the related work section, we have replaced the sentence with the following clearer description: "Most of these approaches rely on the operator tree, which, as noted by Yin et al. (2024), can only approximate the handling of existential first-order logic formulas."
> > >
> > > **For Q5:**
> > >
> > > > Line 182: Is DNF really scalable for handling disjunction operators? I feel the opposite.
> > >
> > > According to the discussion in [10], answering a disjunctive query based on DNF is equivalent to answering $N$ conjunctive queries. In practice, $N$ might not be so large, and all the $N$ computations can be parallelized. In this case, the complexity of solving a disjunctive query depends on the complexity of solving conjunctive queries. Therefore, DNF is scalable in practical applications. A recent work [11] uses DNF to handle disjunction operations on massive knowledge graphs, which also indirectly verifies the scalability of the DNF-based approach.
> > >
> > >
> > > **For Q6:**
> > >
> > > > Equation 9 & 10: What if we get negative values after subtraction?
> > >
> > > With the use of the normalization operation, any negative values resulting from subtraction will not affect the subsequent message-passing process. The normalization operation ensures that each fuzzy vector remains valid.
> > >
> > >
> > > **For Typos:**
> > >
> > > Thanks for your suggestions, we have corrected these errors in the revised paper.
> > >
> > >
> > >
> > > [1] Logical message passing networks with one-hop inference on atomic formulas. ICLR 2023
> > >
> > > [2] Conditional logical message passing transformer for complex query answering. KDD 2024
> > >
> > > [3] Faithful Embeddings for Knowledge Base Queries. Sun et al. NeurIPS 2020
> > >
> > > [4] Complex query answering with neural link predictors. ICLR 2021
> > >
> > > [5] Neural-symbolic entangled framework for complex query answering. NeurIPS 2022
> > >
> > > [6] Answering Complex Logical Queries on Knowledge Graphs via Query Computation Tree Optimization. ICML 2023
> > >
> > > [7] Adapting neural link predictors for data-efficient complex query answering. NeurIPS 2023
> > >
> > > [8] Rethinking complex queries on knowledge graphs with neural link predictors. ICLR 2024
> > >
> > > [9] Neural-Symbolic Models for Logical Queries on Knowledge Graphs. ICML 2022
> > >
> > > [10] Reasoning over knowledge graphs in vector space using box embeddings. ICLR 2020
> > >
> > > [11] Smore: Knowledge graph completion and multi-hop reasoning in massive knowledge graphs. KDD 2022

---

> > > > ### Comment · Reviewer_fVNG · 2024-11-25
> > > >
> > > > Thanks the authors for their response. Unfortunately I can't agree with several points in their response.
> > > >
> > > > My main concern is about the contribution of this paper. The authors only discussed what they did in the paper, but seemed to intentionally avoid comparison against the literature of complex queries. The contribution of this paper is like a mixture of LMPNN, EmQL and CQD with some minor engineering modifications. In their response, the authors claimed that their one-hop inference and message passing is different from LMPNN, but the difference only lies in the instantiation of LMPNN -- with neural and symbolic components that are similar to EmQL. Again, the authors claimed the difference between their neural-symbolic design and EmQL as logical message passing and no additional training, which are copied from LMPNN and CQD respectively. The authors acknowledged that the fuzzy logic they used shares many spirits with CQD, similar to many other methods in the literature. This aligns with my point that fuzzy logic and its interpretability are widely known in the community of complex queries. They can't be regarded as significant contributions of this paper.
> > > >
> > > > I appreciate the interpretability examples and hyperparameter sensitivity results the authors provided in their response. However, they only address some of my minor concerns. Hence I decide to keep my score.

---

> > > > > ### Author Response · Authors · 2024-11-26
> > > > >
> > > > > Thanks for your reply. We hope the following response clears up your misunderstandings.
> > > > >
> > > > > > The authors only discussed what they did in the paper, but seemed to intentionally avoid comparison against the literature of complex queries. The contribution of this paper is like a mixture of LMPNN, EmQL and CQD with some minor engineering modifications.
> > > > >
> > > > > We did not deliberately avoid comparing LMPNN, EmQL, and CQD as you mentioned. There is no reason for us to do so because NSMP is significantly different from these models and performs notably better. Apart from using the neural one-hop inference function of LMPNN and the t-norm fuzzy logic also utilized in CQD, NSMP has no commonalities with LMPNN and CQD. We will discuss their differences in detail below. Regarding EmQL, we have previously discussed that NSMP and EmQL are fundamentally different. We believe that you may have some misunderstandings about our contributions; otherwise, we find it difficult to comprehend your statement: ```some minor engineering modifications```. We believe we have clearly expressed our contributions and novel designs in our revised paper and rebuttal. However, for any remaining misunderstandings, we will consider further discussing these aspects in depth below.
> > > > >
> > > > > > In their response, the authors claimed that their one-hop inference and message passing is different from LMPNN, but the difference only lies in the instantiation of LMPNN -- with neural and symbolic components that are similar to EmQL.
> > > > >
> > > > > As mentioned in my previous rebuttal, the only similarity between NSMP and LMPNN lies in the introduction of LMPNN's neural message encoding function, which we have included as background knowledge in Section 3 of our paper. All other components of NSMP, including neural-symbolic one-hop inference, message passing with dynamic pruning, and the node state update based on fuzzy logic, are unrelated to LMPNN. Overall, NSMP differs from previous message passing methods, including LMPNN, in three fundamental aspects, which underpin its main contributions. These distinctions are: (1) symbolically integrated message computation (Section 4.1 in the revised paper), (2) a non-parametric node state update scheme based on fuzzy logic (Section 4.2.2 in the revised paper), and (3) a dynamic pruning strategy (Section 4.2.1 in the revised paper). Therefore, regarding your statement that ```the difference only lies in the instantiation of LMPNN -- with neural and symbolic components that are similar to EmQL```, we believe there is a misunderstanding.
> > > > >
> > > > > > Again, the authors claimed the difference between their neural-symbolic design and EmQL as logical message passing and no additional training, which are copied from LMPNN and CQD respectively. The authors acknowledged that the fuzzy logic they used shares many spirits with CQD, similar to many other methods in the literature. This aligns with my point that fuzzy logic and its interpretability are widely known in the community of complex queries. They can't be regarded as significant contributions of this paper.
> > > > >
> > > > > We believe that the above discussion has clearly conveyed the differences between NSMP and LMPNN's logical message passing mechanism. Regarding your statement ```copied from```, we believe there is a misunderstanding. Now, let's discuss the differences between NSMP and CQD. The only commonality between NSMP and CQD is the use of t-norm fuzzy logic, which is employed in most neural-symbolic CQA methods. CQD is a step-by-step method based on beam search or continuous optimization, whereas NSMP is a message-passing-based method. From a methodological perspective, the two are fundamentally different. Regarding your statement ```They can't be regarded as significant contributions of this paper```, there may be a misunderstanding here. In the Introduction of our paper, we pointed out that limited interpretability and the need for training on complex queries are among the limitations of existing message passing CQA methods. This means that our contribution is discussed within the scope of message passing CQA methods. Specifically, by introducing symbolic reasoning and fuzzy logic, we enable message passing CQA methods to answer arbitrary $\text{EFO}_1$ queries without requiring training on complex queries while providing interpretability, similar to other step-by-step methods. In particular, the efficiency advantages of the message passing method compared to step-by-step methods have been discussed in detail in Section 4.3 of our revised paper. Therefore, our core contribution lies in enabling an efficient message passing method to achieve strong performance on complex queries without additional training and to provide interpretability.

---

### Official Review · Reviewer_tbFB · 2024-11-04

**Soundness:** 2
**Presentation:** 3
**Contribution:** 3
**Rating:** 5
**Confidence:** 3

**Summary:**

This paper discusses the neural symbolic approach to address the complex query answering over knowledge graphs by leveraging pretrained link predictor in a neuro-symbolic way. The proposed neuro-symbolic framework follows the message passing GNN framework. The key technique is to formulate the message passing and answer ranking module into the ensembles of both neural update (following previous logical message passing) and symbolic updates (following the fuzzy logic). The performance in many datasets for various query types demonstrated the effectiveness of the models. Notably, the framework is shown to be very efficient than previous fuzzy logic based methods.

**Strengths:**

1. The presentation is easy to follow.
2. The overall framework is somewhat novel and empirically significant.
3. the experiments is comprehensive on both different underlying knowledge graph and different query types.

**Weaknesses:**

My particular concern for this paper is whether it can achieves the real efficiency over the fuzzy logic inference approach, such as QTO and FIT. Because the updates on the internal states also includes an adjacency matrix $M_r$, which is $O(n^2)$ space and the fuzzy vector of $O(n)$ space, where $n=|V|$ is the cardinality of the entity set. The calculation of the neuro-symbolic message is of the same cost as (sparse) matrix multiplication, which is already at least quadratic. In that sense, I cannot see the reason in Figure 3, that the speed up of NSMP over FIT, which suggested at least 10 times speed up in NELL.

**Questions:**

1. Could you please explain the complexity of FIT and your algorithm on both acyclic and cyclic query? in terms of the size of graph and the size of query.
2. Please address my weakness.
3. For the dynamic pruning of the message, I would like to know whether the total message to compute and total states to update can be significantly better than the original LMPNN without pruning, say $O(L m) = O(m^2)$, where $L$ is the total number of layers of message and $O(m)$ is the number of predicates in the query graph.

---

> ### Author Response · Authors · 2024-11-21
> **(1/2)**
>
> Thanks for your thoughtful comments. We hope the following responses address your concerns.
>
> **For weaknesses, Q1 and Q2:**
>
>
> According to [1], QTO [2] is a simplified version of FIT [1]. That is, FIT is a natural extension of QTO. Therefore, here we focus on the computational complexity of NSMP compared to the state-of-the-art step-by-step neural-symbolic method FIT to reveal why the message-passing-based method can provide more efficient inference than the step-by-step approach. In addition, we have also added a detailed discussion on computational complexity in Section 4.3 of the revised paper we uploaded.
>
>
> * **Space Complexity**. For the space complexity of neural components, both NSMP and FIT use a pre-trained neural link predictor with a complexity of $\mathcal{O}((|\mathcal{V}|+|\mathcal{R}|)d)$, where $d$ is the embedding dimension. The symbolic one-hop inference component of NSMP utilizes relational adjacency matrices, which contain $|\mathcal{R}|\cdot |\mathcal{V}|^2$ entries. However, due to the sparsity of KG, most entries are $0$. With the help of sparse matrix techniques, the adjacency matrices can be stored efficiently. In this regard, NSMP has a space complexity similar to that of FIT. The neural adjacency matrix used in FIT also contains $|\mathcal{R}|\cdot |\mathcal{V}|^2$ entries and can be efficiently stored by setting appropriate thresholds.
>
> * **Time Complexity**. Since the computational bottleneck of NSMP lies in the symbolic-related parts, we focus on discussing the time complexity of this aspect. Due to the sparsity of the adjacency matrix and fuzzy vector, both NSMP and FIT can utilize sparse techniques for efficient inference. But for simplicity, we will not consider this sparsity in the following discussion.
>
>
>   According to Equations 7-10, the complexity of symbolic one-hop inference is $\mathcal{O}(|\mathcal{V}|^2)$, so the complexity of neural-symbolic message encoding function $\varrho$ is approximately $\mathcal{O}(|\mathcal{V}|^2)$. This means that the complexity of message computation during message passing is linear to $\mathcal{O}(|\mathcal{V}|^2)$. For the node state update process, symbolic state update and neural state update are involved, corresponding to Equations 17 and 18, respectively. The complexity of the symbolic node state update is linear to $\mathcal{O}(|\mathcal{V}|)$, while the complexity of the neural node state update is $\mathcal{O}(|\mathcal{V}|d)$, where $d$ is the embedding dimension. Since we have $d\ll |\mathcal{V}|$, the total computational complexity of NSMP is approximately linear to $\mathcal{O}(|\mathcal{V}|^2)$. In particular, as a message-passing-based approach, the solving process of NSMP is the same for both acyclic and cyclic queries, i.e., performing message passing on the query graph. Consequently, the complexity of NSMP for any $\text{EFO}_1$ formula is approximately linear to $\mathcal{O}(|\mathcal{V}|^2)$.
>
>   For the previous step-by-step neural-symbolic method FIT, according to [1], FIT solves the acyclic query by continuously removing constants and leaf nodes, and the complexity of this process is approximately linear to $\mathcal{O}(|\mathcal{V}|^2)$. However, for cyclic queries, FIT needs to enumerate one variable within the cycle as a constant node, so the complexity is $\mathcal{O}(|\mathcal{V}|^n)$, where $n$ is the number of variables in the query graph. In contrast, the complexity of NSMP on cyclic queries is approximately linear to $\mathcal{O}(|\mathcal{V}|^2)$, which means that NSMP can provide more efficient inference on cyclic queries. As we show in Section 5.2 in the revised paper, NSMP achieves faster inference times when compared with FIT on cyclic queries, with speedup ranging from 69$\times$ to over 150$\times$. Moreover, while both NSMP and FIT exhibit a complexity linear to $\mathcal{O}(|\mathcal{V}|^2)$ for acyclic queries, the computation of different messages in NSMP's message-passing process is independent. This independence enables a parallelized computation of messages. In contrast, FIT requires removing constant and leaf nodes step by step, where the steps are interdependent, necessitating a serial process. As a result, even for acyclic queries, the message-passing-based NSMP can achieve more efficient inference. As demonstrated in Section 5.2 in the revised paper, NSMP achieves a speedup of at least 10$\times$ for acyclic queries in NELL995.
>
> Due to the space limitation, please see the following comment, i.e., (2/2).

---

> > ### Author Response · Authors · 2024-11-21
> > **(2/2)**
> >
> > **For Q3:**
> >
> > We may not have understood your question correctly. If we have misunderstood your question and have not addressed it, please point it out. The dynamic pruning strategy can effectively reduce the computation of messages and state updates. Compared with the original LMPNN [3], the conditional message passing mechanism proposed by CLMPT [4] has been able to effectively reduce a part of the message computations and state update computations. CLMPT does not consider passing messages to constant nodes, nor does it update the states of constant nodes, thereby reducing this part of the computations. The dynamic pruning proposed in our paper is built on conditional message passing. In addition to pruning the noise messages between constants and variables like CLMPT, we also dynamically prune unnecessary noise messages between variable nodes and do not update the states of variable nodes that have not received any messages. Therefore, compared with the bidirectional unconditional message passing in the original LMPNN, our proposed dynamic pruning strategy can effectively reduce the computation of unnecessary messages and state updates.
> >
> >
> >
> > [1] Rethinking complex queries on knowledge graphs with neural link predictors. ICLR 2024
> >
> > [2] Answering Complex Logical Queries on Knowledge Graphs via Query Computation Tree Optimization. ICML 2023
> >
> > [3] Logical message passing networks with one-hop inference on atomic formulas. ICLR 2023
> >
> > [4] Conditional logical message passing transformer for complex query answering. KDD 2024

---

> > > ### Comment · Reviewer_tbFB · 2024-11-28
> > > **Reply to the rebuttal.**
> > >
> > > Dear authors,
> > >
> > > Sorry for the late reply. After looking into the responses, I am still confused with your results about the efficiency, in particular, Table 8.
> > >
> > > As you stated, FIT and NSMP follow the same complexity in each step for the acyclic query. So, is it okay for me to think there are no differences in terms of time complexity between one FIT step of removing leaf nodes and one symbolic message calculation in NSMP?
> > >
> > > If the answer is yes, the explanation of the difference between NSMP and FIT in efficiency relates to my third question. More specifically, either parallelization or dynamic pruning plays an important role for the acceleration.
> > >
> > > For the parallelization, I think at least the number of layers of MPNNs (LMPNN or NSMP), say $L$, is the same as the diameter of the query graph, it is of the same order of number of FIT steps of removing leaf nodes.
> > >
> > > For dynamic pruning, I think the strategy in [4] are very strange to lead to significant acceleration of surprisingly tens of times because the cost of each step is the same, but the number of steps can only be compressed in a small constant ratio.
> > >
> > > Please help me to figure out where is the gap between my understanding and your implementation.
> > >
> > > Best.

---

> > > > ### Author Response · Authors · 2024-11-29
> > > >
> > > > Thank you for your insightful comments. We hope the following responses address your concerns.
> > > >
> > > > Our previous rebuttal simplified the time complexity analysis by not considering sparsity. In fact, sparsity helps accelerate computation in both FIT and NSMP. However, there are differences in sparsity between them, which results in NSMP offering more efficient computation.
> > > >
> > > > The results in Table 8 of our paper present the inference time of NSMP and FIT on each query type on FIT datasets. One important characteristic of FIT datasets is that all queries include the existential variable. In this case, according to the analysis in FIT, FIT and QTO are equivalent on these acyclic queries containingexistential variables. Therefore, we now focus on analyzing the time complexity of NSMP and QTO on these acyclic queries while considering sparsity. According to the analysis in QTO, due to the sparsity of the KG, the time complexity of QTO is actually $\mathcal{O}(T^*(v_k) \cdot \frac{L_1}{|\mathcal{V}|})$, where $T^*(v_k)$ is a sparse vector, $L_1$ is the total number of non-zero elements in the sparse neural adjacency matrix, and $\frac{L_1}{|\mathcal{V}|}$ can be interpreted as the average number of non-zero elements per row in this matrix.
> > > >
> > > > For NSMP, the neural-symbolic message computation is essentially the sparse matrix multiplication between the fuzzy vector (or one-hot vector) and the relational adjacency matrix. In our implementation, we use a thresholded normalization function. In this case, the fuzzy vector is also sparse. The time complexity of the message computation is $\mathcal{O}(K \cdot \frac{L_2}{|\mathcal{V}|})$, where $K$ is the number of non-zero elements in the sparse fuzzy vector, and $L_2$ is the total number of non-zero elements in the sparse relational adjacency matrix. The sparsity of the fuzzy vector in NSMP and the vector $T^*(v_k)$ in QTO are both determined by the chosen threshold. Therefore, we can assume that with appropriately set hyperparameters, $K = |T^*(v_k)|$. For $L_1$, the neural adjacency matrix in QTO can be sparsified by setting an appropriate threshold such that $L_1 < |\mathcal{V}|^2$. However, the relational adjacency matrix used in NSMP only contains $0$ and $1$. In theory, the neural adjacency matrix in QTO only achieves the same sparsity as the relational adjacency matrix used in NSMP when the threshold is set to $1$. However, setting the threshold to $1$ in QTO would make no sense, as it would strip the model of its neural reasoning capability. This suggests that, in terms of the adjacency matrix, NSMP theoretically has a more efficient computational complexity compared to QTO, with $\frac{L_2}{|\mathcal{V}|} < \frac{L_1}{|\mathcal{V}|} < |\mathcal{V}|$. Thus, when considering sparsity, NSMP has a better computational complexity for acyclic queries in FIT datasets compared to FIT and QTO.
> > > >
> > > > Regarding the efficiency gained from our proposed dynamic pruning strategy, we have previously discussed this in our response to Reviewer ThD9. Specifically, we provided an empirical result showing that the dynamic pruning strategy reduces the average inference time by 23.8%. However, this level of acceleration is clearly not the primary reason for NSMP's significantly better inference efficiency compared to FIT. The main reason for NSMP's faster inference time relative to FIT is that NSMP has a better computational complexity. The dynamic pruning strategy just further enhances the efficiency of the already efficient message-passing mechanism.
> > > >
> > > > In summary, we believe that the superior computational efficiency of NSMP compared to FIT has been adequately explained. We will include the above discussion in the revised paper.

---

### Official Review · Reviewer_ThD9 · 2024-11-04

**Soundness:** 3
**Presentation:** 3
**Contribution:** 2
**Rating:** 6
**Confidence:** 4

**Summary:**

The research proposes a novel method for answering Complex Queries within knowledge graphs ($\text{EFO}_1$) through the use of Neura-symbolic  Message Passing with neural link prediction and fuzzy set theory for aggregation. The method is claimed to be computationally efficient as is does not require training on complex queries and can simply utilise a pre-trained link prediction on a given Knowledge graph. The method also allows dynamic pruning that filters the noise from the initial, un-updated layers during the message-passing process. The method shows competitive results on a set of Benchmarks from BetaE and FIT and provides ablations showcasing the need for dynamic pruning and impact of the hyperparameters.

**Strengths:**

The method suggests a novel method for answering Complex Queries($\text{EFO}_1$) over knowledge graphs that uses pretrained link predictors with Neura-symbolic message passing and fuzzy set theory for aggregation. The method allows the encoding of both local and global information along with both neural and symbolic representations during the message-passing process. This is coupled with an interesting dynamic pruning technique that filters out the impact/noise from the initial layers of un-updated variable nodes. The method is rather competitive on a set of benchmarks from BetaE and FIT and allows for more interpretable reasoning by analysing the fuzzy intermediate representations.

**Weaknesses:**

1. While the method is ripe with novel ideas I feel that the benefits it brings have already been explored in prior research. Methods like and stemming from CQD and CQD-A train only a single link predictor, thus circumventing the need to train on complex queries. They, along with GNN-QE and others, offer interpretable query answers by exposing the intermediate answers (top-k). The use of fuzzy logic is also explored within these papers. We see that for example, in NELL (both BetaE and FIT versions), other methods are only marginally different (CQD-A) or are on average, better (FIT) than the suggested framework. Can you please elaborate on the novel additions in NSMP and what do they exactly contribute?  Can you elaborate on the comparison in the amount of parameters w.r.t. other benchmarks are they comparable?

2. The novel idea of dynamic pruning is rather interesting; however, the ablation shows a sizable average increase only for queries sampled in FIT. Can you explain the reason for this? How much efficiency does dynamic pruning add to the overall network in terms of inference?

2.5. As direct link prediction is chosen along a product T-norm for fuzzy aggregation, the method is bound to have intermediate answers that do not interact together as outlined in CQD-A. This means that because the probability ranges of different intermediate answers are not homogenous aggregating with a product t-norm can cause massive discrepancies during message propagation. Has any analysis of intermediate answers been completed?

3. The choice not to include "pni" is not justified, as Yin et al. (2024) includes that scheme with an $\text{EFO}_1$ definition (Example 10 and Property 12 in the paper) and just resamples w.r.t. their definition. The original version of "pni" proposed in BetaE is simply the universal quantifier version that is resampled in FIT w.r.t. their definition. Can you elaborate on the reasoning to not include "pni" ? Has it been tested on ?

4. The paper cited as the reason to not include FB15K (Toutanova & Chen, 2015)) does not claim test leakage but rather proposes a reconstruction of FB15 with the removal of "near-duplicate" or "inverse-duplicate" relations. While FB 15L does have some discrepancies bot BetaE  and FIT still include it in dataset construction and most of the CQA methods are benchmarked on it. Can you please elaborate on the choice of not using FB15K and if there is a stronger argument against the use of FB15K?

**Questions:**

**Question set Q1: For context, see Point 1 in weaknesses:**

*Can you please elaborate on the novel additions in NSMP and what do they exactly contribute?*

*Can you elaborate on the comparison in the amount of parameters w.r.t. other benchmarks are they comparable?*

**Question set Q2: For context, see Point 2 in weaknesses:**

*Can you explain the reason for this?*

*How much efficiency does dynamic pruning add to the overall network in terms of inference?*

**Question set Q2.5: For context, see Point 2.5 in weaknesses:**

*Has any analysis of intermediate answers been completed?*

**Question set Q3: For context, see Point 3 in weaknesses:**

*Can you elaborate on the reasoning to not include "pni" ?*

*Has it (pni) been tested on ?*

**Question set Q4: For context, see Point 4 in weaknesses:**

*Can you please elaborate on the choice and if there is a stronger argument against the use of FB15K ?*

---

> ### Author Response · Authors · 2024-11-21
> **(1/3)**
>
> Thanks for your insightful comments. We hope the following responses address your concerns.
>
> **For W1 and Q1:**
>
> > We see that for example, in NELL (both BetaE and FIT versions), other methods are only marginally different (CQD-A) or are on average, better (FIT) than the suggested framework.
>
> Although NSMP exhibits only marginally different performance compared to $\text{CQD}^{\mathcal{A}}$ [1] on NELL995, it does not require any training on complex queries, whereas $\text{CQD}^{\mathcal{A}}$ relies on training for such queries, even though it is data-efficient. In the absence of training data, $\text{CQD}^{\mathcal{A}}$ degenerates to its original version, CQD [2], which NSMP significantly outperforms. Therefore, NSMP can be considered superior to $\text{CQD}^{\mathcal{A}}$ overall. As for FIT [3], while NSMP achieves second-best results compared to FIT, it offers more efficient inference while maintaining competitive performance. Specifically, NSMP achieves a speedup of at least 10$\times$ for complex queries on NELL995.
>
> > Can you please elaborate on the novel additions in NSMP and what do they exactly contribute?
>
>
> The core contribution of NSMP lies in enabling an efficient message passing CQA method to generalize to arbitrary $\text{EFO}_1$ queries without requiring training on complex queries, while also providing interpretability. In general, our contributions can be summarized as follows:
> * We propose a neural-symbolic message passing framework that, for the first time, integrates neural and symbolic reasoning within a message passing CQA model. By leveraging fuzzy logic theory, our approach can naturally answer arbitrary existential first order logic formulas without requiring training on complex queries while providing interpretability.
> * We propose a novel dynamic pruning strategy to filter out unnecessary noisy messages between variable nodes during message passing, thereby reducing unnecessary computation and improving performance.
> * Extensive experimental results on benchmark datasets show that NSMP achieves a strong performance. In particular, NSMP significantly improves the performance of message passing CQA models on negative queries by introducing symbolic reasoning and fuzzy logic.
> * Through computational complexity analyses, we reveal that our message-passing-based method can provide more efficient inference than the current state-of-the-art step-by-step neural-symbolic method, and empirically verify this.
>
> We uploaded a revised PDF where we clarify the contributions mentioned above in the Introduction section.
>
> > Can you elaborate on the comparison in the amount of parameters w.r.t. other benchmarks are they comparable?
>
> We have added a discussion on computational complexity in Section 4.3 of the revised paper, which includes an analysis of space complexity, including the number of parameters. We focus on comparing our approach to the state-of-the-art step-by-step neural-symbolic method, FIT. Through this discussion, we reveal why our message-passing-based method enables more efficient inference compared to the step-by-step FIT approach, and we empirically validate this claim. Overall, Section 4.3 demonstrates that NSMP and FIT have similar space complexity and can greatly reduce the number of parameters through sparse techniques. Regarding time complexity, we highlight the computational advantages of the message passing mechanism, as detailed in Section 4.3 of the revised paper.
>
> Due to the space limitation, please see the following comment, i.e., (2/3).

---

> > ### Author Response · Authors · 2024-11-21
> > **(2/3)**
> >
> > **For W2 and Q2:**
> >
> > > however, the ablation shows a sizable average increase only for queries sampled in FIT. Can you explain the reason for this?
> >
> > The proposed dynamic pruning strategy is designed to filter out unnecessary noisy messages between variable nodes. Consequently, the number of variable nodes and their interactions in a query graph significantly influence the performance gains achievable through dynamic pruning. As illustrated in Figure 4 and Figure 5 of the manuscript, the query types in the FIT datasets involve more variable nodes compared to those in the BetaE [4] datasets. Moreover, the interactions between variable nodes are also more frequent in FIT queries, such as in "3c" and "3cm" queries. As a result, during message passing, query types in FIT datasets exhibit more noisy messages between variable nodes than in BetaE datasets. This explains why the dynamic pruning strategy achieves relatively higher performance gains on the FIT datasets.
> >
> >
> > > How much efficiency does dynamic pruning add to the overall network in terms of inference?
> >
> >
> > The dynamic pruning strategy filters out noisy messages between variable nodes, thereby reducing the computational cost associated with these messages. As discussed in Section 4.3 of the revised paper, the primary computational bottleneck of NSMP lies in the computation of neural-symbolic messages. This indicates that the dynamic pruning strategy effectively alleviates a portion of the computational burden during message passing. Additionally, the dynamic pruning strategy does not update the states of variable nodes that do not receive any messages, further reducing the computation required for state updates. In summary, the dynamic pruning strategy can reduce the state update computation and message computation in the forward process, effectively improving inference efficiency.
> >
> >
> > **For W2.5 and Q2.5:**
> >
> > > Has any analysis of intermediate answers been completed?
> >
> > In the revised paper, we have included a case study (see Appendix G) analyzing the interpretability of NSMP, focusing on the fuzzy sets associated with different variables. Although, as you noted, the various intermediate states of a node in NSMP's message-passing process are not homogenous, the case study and main experimental results demonstrate that directly applying product t-norm aggregation to these intermediate states can still yield effective results. The most direct evidence is that the parameter-free NSMP outperforms $\text{CQD}^{\mathcal{A}}$, which requires training on complex queries, even though the intermediate states in $\text{CQD}^{\mathcal{A}}$ interact together. Therefore, we believe directly aggregating a variable node's heterogeneous intermediate states using the product t-norm is a practical approach. This approach also enables NSMP to achieve more efficient inference on acyclic queries compared to step-by-step FIT (see Section 4.3 for details).
> >
> >
> >
> > **For W3 and Q3:**
> >
> >
> > > Can you elaborate on the reasoning to not include "pni" ? Has it been tested on ?
> >
> >
> > We believe there may be a misunderstanding. As stated in our paper, we follow the evaluation benchmark of FIT. We evaluate the "pni" query on the FIT datasets. This means that we did not exclude the "pni" query. The test results of the "pni" query are shown in Table 2 of our paper. According to FIT, the "pni" query in the BetaE datasets is answered as the universal quantifier version. FIT maintains the "pni" query type but re-samples the answers according to their own definition. Therefore, we follow FIT and only evaluate the "pni" query with re-sampled answers, as defined in the FIT datasets. The FIT datasets we used are the same as those proposed in FIT.
> >
> > Due to the space limitation, please see the following comment, i.e., (3/3).

---

> > > ### Author Response · Authors · 2024-11-21
> > > **(3/3)**
> > >
> > > **For W4 and Q4:**
> > >
> > > > Can you please elaborate on the choice of not using FB15K and if there is a stronger argument against the use of FB15K?
> > >
> > > We follow [5,6] and exclude FB15k. As stated in [5]: ```We exclude FB15k (Bordes et al. 2013) since this dataset suffers from major test leakage (Toutanova and Chen 2015).``` In addition, a similar viewpoint is also expressed in [7]. Many triples in FB15k are inverses that cause leakage from the training to testing and validation splits. Therefore, Toutanova and Chen [8] created FB15k-237 to ensure that the testing and evaluation datasets do not have inverse relation test leakage. A similar viewpoint has been recognized by multiple studies [9,10]. For example, as noted in [9]: ```FB15k-237 is a subset of FB15k built by Toutanova and Chen, inspired by the observation that FB15k suffers from test leakage, consisting in test data being seen by models at training time. In FB15k this issue is due to the presence of relations that are near-identical or the inverse of one another. In order to assess the severity of this problem, Toutanova and Chen have shown that a simple model based on observable features can easily reach state-of-the-art performance on FB15k. ```
> > >
> > >
> > > [1] Adapting neural link predictors for data-efficient complex query answering. NeurIPS 2023
> > >
> > > [2] Complex query answering with neural link predictors. ICLR 2021
> > >
> > > [3] Rethinking complex queries on knowledge graphs with neural link predictors. ICLR 2024
> > >
> > > [4] Beta embeddings for multi-hop logical reasoning in knowledge graphs. NeurIPS 2020
> > >
> > > [5] Fuzzy logic based logical query answering on knowledge graphs. AAAI 2022
> > >
> > > [6] Neural-symbolic entangled framework for complex query answering. NeurIPS 2022
> > >
> > > [7] Mask and reason: Pre-training knowledge graph transformers for complex logical queries. KDD 2022
> > >
> > > [8] Observed versus latent features for knowledge base and text inference. Workshop on CVSC in ACL 2015
> > >
> > > [9] Knowledge Graph Embedding for Link Prediction: A Comparative Analysis. TKDD 2021
> > >
> > > [10] CODEX: A Comprehensive Knowledge Graph Completion Benchmark. EMNLP 2020

---

> > > ### Comment · Reviewer_ThD9 · 2024-11-24
> > > **Reply to Part 2 of the author comments**
> > >
> > > 1. ```How much efficiency does dynamic pruning add to the overall network in terms of inference?```
> > >
> > > When asking the question, the reviewer expected to see either theoretical or empirical results on how much computational efficiency, on average, the dynamic pruning adds to the framework. Can the authors please add a theoretical (mathematical complexity analysis) or empirical argument to their prior answer?
> > >
> > > All the other questions in this part have been adequately answered for which the reviewer thanks the authors.

---

> > > > ### Author Response · Authors · 2024-11-25
> > > > **Response to Newly Raised Questions (Part 1)**
> > > >
> > > > Thanks for your comments. We hope the following responses address your outstanding questions.
> > > >
> > > > **For newly raised Q1:**
> > > >
> > > > We can demonstrate the superiority of NSMP over $\text{CQD}^{\mathcal{A}}$ using the two data-efficient variants of $\text{CQD}^{\mathcal{A}}$ you mentioned. For these two data-efficient variants, their performance is not, as you stated, nearly equivalent to that of the original $\text{CQD}^{\mathcal{A}}$. On the contrary, their performance degrades significantly compared to the original $\text{CQD}^{\mathcal{A}}$. The results of $\text{CQD}^{\mathcal{A}}$ compared in our paper are derived from the original paper [1]. The authors of $\text{CQD}^{\mathcal{A}}$ obtained these results by training on the dataset that includes all 2i, 3i, 2in, and 3in queries. However, as you suggested, to fairly compare $\text{CQD}^{\mathcal{A}}$ and NSMP, we should compare $\text{CQD}^{\mathcal{A}}$ using the two data-efficient variants proposed in the original paper [1] (i.e., "FB237, 1%" and "FB237, 2i, 1%") since NSMP does not have trainable parameters. In this case, as shown in the table below, we have verified the superiority of NSMP over $\text{CQD}^{\mathcal{A}}$.
> > > >
> > > > | Model                         | 1p   | 2p   | 3p   | 2i   | 3i   | pi   | ip   | 2u   | up   | AVG.(P) | 2in  | 3in  | inp  | pin | AVG.(N) |
> > > > | ----------------------------- | ---- | ---- | ---- | ---- | ---- | ---- | ---- | ---- | ---- | ------- | ---- | ---- | ---- | --- | ------- |
> > > > | CQD-A(FB237,2i,3i,2in,3in)    | **46.7** | 13.6 | 11.4 | 34.5 | 48.3 | 27.4 | 20.9 | **17.6** | 11.4 | 25.7    | **13.6** | 16.8 | 7.9  | **8.9** | 11.8    |
> > > > | CQD-A(FB237,1%)               | **46.7** | 11.8 | 11.4 | 33.6 | 41.2 | 24.8 | 17.8 | 16.5 | 8.7  | 23.6    | 10.8 | 13.9 | 5.9  | 5.4 | 9.0     |
> > > > | CQD-A(FB237,2i,1%)            | **46.7** | 11.8 | 11.2 | 30.4 | 40.8 | 23.4 | 18.3 | 15.9 | 9.0  | 23.1    | 9.4  | 10.3 | 5.2  | 4.5 | 7.4      |
> > > > | NSMP(No trainable Parameters) | **46.7** | **15.1** | **12.3** | **38.7** | **52.2** | **31.2** | **23.3** | 17.2 | **11.9** | **27.6**    | 11.9 | **17.6** | **10.8** | 7.9 | **12.0**     |
> > > >
> > > > From the aforementioned results, in the case where both of these variants still require training data (though very data-efficient), the NSMP, which has no trainable parameters, improves the average MRR score on EPFO queries by **16.9%** and **19.5%** respectively compared to these two variants. On negative queries, the average MRR score is improved by **33.3%** and **62.2%**, respectively. Considering such significant performance improvements, we believe that the superiority of NSMP over $\text{CQD}^{\mathcal{A}}$ has been demonstrated.
> > > >
> > > > **For newly raised Q2:**
> > > >
> > > > > Can you also provide the statistics w.r.t. the speedups over CQD-A if such exist?
> > > >
> > > > We are unable to find the code of $\text{CQD}^{\mathcal{A}}$, and thus cannot directly measure the inference time of $\text{CQD}^{\mathcal{A}}$. However, we can discuss the efficiency of NSMP compared to $\text{CQD}^{\mathcal{A}}$ from an indirect perspective. Considering that the main improvement of $\text{CQD}^{\mathcal{A}}$ over the original CQD lies in the introduction of an additional trainable calibration module, CQD is expected to have slightly faster inference speeds compared to $\text{CQD}^{\mathcal{A}}$. Let us analyze CQD. Its time complexity is $\mathcal{O}(N'|\mathcal{V}|bd)$, where $N'$ is the number of predicates in the query, $d$ is the embedding dimension, and $b$ is the beam size. According to the analyses in QTO [2], while the worst-case time complexity of QTO is $\mathcal{O}(N'|\mathcal{V}|^2)$, the sparsity of the KG makes the actual time complexity approximately $\mathcal{O}(N'|\mathcal{V}|\cdot \max_k |T^*(v_k) > 0|)$, where $T^*(v_k)$ is a sparse vector. Due to this sparsity and the acceleration of matrix computations, QTO has been empirically shown in [2] to provide more efficient inference than CQD, implying that QTO should also be more efficient than $\text{CQD}^{\mathcal{A}}$. As demonstrated in [3], FIT is a natural extension of QTO, which indicates that FIT can deliver more efficient inference compared to $\text{CQD}^{\mathcal{A}}$. Combining this with our computational complexity analyses of NSMP and FIT in Section 4.3, along with the empirical results, we conclude that NSMP can provide more efficient inference than $\text{CQD}^{\mathcal{A}}$.
> > > >
> > > > Due to the space limitation, please see the following comment, i.e., Response to Newly Raised Questions (Part 2).

---

> > > > > ### Author Response · Authors · 2024-11-25
> > > > > **Response to Newly Raised Questions (Part 2)**
> > > > >
> > > > > **For newly raised Q3:**
> > > > >
> > > > > > Thanks for the comparison with FIT, however FIT is not necessarily parameter-efficient. Can you elaborate on the comparison in the amount of parameters w.r.t. CQD-A and GNN-QE are they comparable to NSMP?
> > > > >
> > > > > In Appendix F of the revised paper, we analyze the space complexity of NSMP. The parameters of the neural component in NSMP correspond to those of the neural link predictor, with a complexity of $\mathcal{O}((|\mathcal{V}|+|\mathcal{R}|)d)$. For the symbolic component of NSMP, the relational adjacency matrices contain $|\mathcal{R}|\cdot |\mathcal{V}|^2$ entries. However, due to the sparsity of the KG, the relational adjacency matrices can be efficiently stored. For $\text{CQD}^{\mathcal{A}}$, the additional trainable network introduced compared to the original CQD is parameter-efficient, so the parameters of $\text{CQD}^{\mathcal{A}}$ can be considered similar to that of CQD, which primarily consists of the parameters of the neural link predictor. As for GNN-QE, based on the analyses in [1], the parameters of GNN-QE mainly stems from its projection network, implemented as NBFNet [4], which has fewer parameters than $\text{CQD}^{\mathcal{A}}$. Overall, NSMP has larger parameters compared to both $\text{CQD}^{\mathcal{A}}$ and GNN-QE. However, $\text{CQD}^{\mathcal{A}}$ and GNN-QE require complex query training data, and their performance degrades significantly in the absence of such data. In contrast, NSMP, which has no trainable parameters and achieves superior performance, is more flexible in adapting to various scenarios.
> > > > >
> > > > > **For newly raised Q4:**
> > > > >
> > > > > We believe there is a misunderstanding. We did not claim to be the first to apply fuzzy logic to message passing CQA methods, but rather the first to integrate neural and symbolic reasoning within message passing CQA methods. To the best of our knowledge, existing message passing CQA methods are purely neural approaches that do not incorporate any symbolic information. Our proposed NSMP is the first symbolic-integrated message passing CQA method.
> > > > >
> > > > > **For newly raised Q5:**
> > > > >
> > > > > > Can the authors please add a theoretical (mathematical complexity analysis) or empirical argument to their prior answer?
> > > > >
> > > > > To verify the previous discussion about the efficiency brought by dynamic pruning (i.e., the previous responses to W2 and Q2), we conducted experiments on inference time on an NVIDIA RTX 3090 GPU. We consider the comparison between the original NSMP (i.e., the version with dynamic pruning) and NSMP without dynamic pruning (i.e., NSMP w/o DP) on FB15k-237. Specifically, we measured the average inference time required by the original NSMP and NSMP w/o DP to process each type of test query on the FIT dataset. The detailed inference times, expressed in milliseconds per query (ms/query), are provided below.
> > > > >
> > > > > | Model       | pni  | 2il  | 3il  | 2m   | 2nm  | 3mp  | 3pm  | im   | 3c   | 3cm  | Avg  |
> > > > > | ----------- | ---- | ---- | ---- | ---- | ---- | ---- | ---- | ---- | ---- | ---- | ---- |
> > > > > | NSMP w/o DP | 14.4 | 11.4 | 14.0 | 16.6 | 16.6 | 29.2 | 29.2 | 19.4 | 32.8 | 39.8 | 22.3 |
> > > > > | NSMP        | 12.8 | 9.0  | 11.8 | 11.8 | 12.0 | 20.6 | 18.8 | 14.8 | 27.0 | 31.8 | 17.0 |
> > > > >
> > > > > From the results above, it can be observed that the dynamic pruning strategy not only effectively enhances model performance (see Ablation Study in the paper) but also improves efficiency. Compared to the variant without dynamic pruning, NSMP achieves a **23.8%** reduction in average inference time. We believe that these empirical results validate our view that the dynamic pruning strategy can effectively enhance efficiency.
> > > > >
> > > > >
> > > > > [1] Adapting neural link predictors for data-efficient complex query answering. NeurIPS 2023
> > > > >
> > > > > [2] Answering Complex Logical Queries on Knowledge Graphs via Query Computation Tree Optimization. ICML 2023
> > > > >
> > > > > [3] Rethinking complex queries on knowledge graphs with neural link predictors. ICLR 2024
> > > > >
> > > > > [4] Neural bellman-ford networks: A general graph neural network framework for link prediction. NeurIPS 2021

---

> > > > > > ### Comment · Reviewer_ThD9 · 2024-11-27
> > > > > >
> > > > > > I thank the authors for going the extra mile to answer the questions. After some deliberation, I decided to increase my score in support of the paper.
> > > > > >
> > > > > > It must be mentioned that some aspects of parameter efficiency and the overall contribution (Answers to Q3 and Q4) still remain dubious.

---

> > ### Comment · Reviewer_ThD9 · 2024-11-24
> > **Reply to Part 1 of the comments**
> >
> > I thank the authors for their detailed response. Although some questions have been cleared in this part I still have other outstanding question.
> >
> > 1. CQD-A has a variation that trains on 1% of some part of the query types or outright 1% of just one type (2i) of a queries that allows to maintain the performance with close to no degradations. As this calibration is highly data efficient, it is a fair comparison to the proposed method. I cannot see a CQD-A comparison on the FIT dataset. Are there any experiments to confirm that NSMP is superior to CQD-A ?
> >
> > 2. Can you also provide the statistics w.r.t. the speedups over CQD-A if such exist?
> >
> > 3. Thanks for the comparison with FIT, however FIT is not necessarily parameter-efficient. Can you elaborate on the comparison in the amount of parameters w.r.t. CQD-A and GNN-QE are they comparable to NSMP?
> >
> > 4. Regarding the contributions I just have a concern with the following point.
> >
> > ```We propose a neural-symbolic message passing framework that, for the first time, integrates neural and symbolic reasoning within a message passing CQA model. By leveraging fuzzy logic theory, our approach can naturally answer arbitrary existential first order logic formulas without requiring training on complex queries while providing interpretability.```
> >
> > I would argue that this is not the first time that fuzzy logic has been applied on top of a message passing method (has been done in numerous papers since CQD, i.e. LMPNN). I agree that the presented method has merit and is novel, but this line does not seem convincing.
> >
> > I would be willing to concede if the comparisons and concerns asked for in questions [1-4] are adressed.

---

### Meta-Review · Area_Chair_78v1 · 2024-12-20

**Metareview:**

This study introduces a neural-symbolic message-passing framework for complex question answering. To mitigate the issue of noisy messages during the message-passing process, a dynamic pruning strategy is proposed.

Two of the reviewers have raised several significant concerns:

- Lack of clear motivation for the method development: The connection between the proposed model components and the existing challenges outlined in the introduction is not well established.
- Limited novelty: Many of the components in the proposed method have been previously proposed in existing studies. Additionally, the authors fail to cite some of them.
- Overlap with prior work: The reviewers pointed significant overlap with some of the previously published paper.

Although the authors made an effort to address these concerns in their rebuttal, the responses were not sufficiently convincing to satisfy the reviewers.

**Additional Comments On Reviewer Discussion:**

No changes. The unsolved points have been summarized in the meta-review.

---

### Decision · Program_Chairs · 2025-01-22

Reject